# The Health System's Response to and the Impact of COVID-19 on Health Services, Providers, and Seekers: A Rapid Review in the Wake of the Pandemic

Ankur Singh Chauhan [1,*], Kultar Singh [2], Rajesh Bhatia [3], Sonalini Khetrapal [4] and Aditya Naskar [2]

1   London School of Hygiene and Tropical Medicine, London WC1E 7HT, UK
2   Sambodhi Research and Communication, Pvt. Ltd., Noida 201301, India; kultar@sambodhi.co.in (K.S.); adityanaskar@hotmail.com (A.N.)
3   Asian Development Bank, New Delhi 110021, India; drrajesh.bhatia1953@gmail.com
4   Asian Development Bank, Mandaluyong, Metro Manila 1550, Philippines; skhetrapal@adb.org
*   Correspondence: ankurheathwalk@gmail.com; Tel.: +44-7824494802

**Abstract:** Background: The COVID-19 pandemic disrupted global healthcare systems, requiring rapid adaptations. This study evaluates the impact on health systems and services in India during the peak of the first wave and its aftermath. It analyses disruptions, adaptive measures, and challenges faced by healthcare providers and seekers to enhance future preparedness. Methods: Primary studies conducted in India exploring the impact of COVID-19 on health services provision, utilisation, availability, and the well-being of providers and seekers were included. Electronic searches were conducted in six databases: PubMed, MEDLINE, Embase, Global Health, CINAHL, and the WHO database on COVID-19. The results were analysed using narrative synthesis. Results and Conclusion: The review examined 38 articles with 22,502 subjects. Health service provision, utilisation, and availability were significantly impacted, particularly in outpatient departments ($n = 19$) and elective services ($n = 16$), while emergency services remained sub-optimal ($n = 20$). Adaptations were made in precautionary measures, protocols, staff allocation, training, personal protective equipment (PPE), infrastructure, and resources. Providers faced mental health challenges including depression, stress ($n = 14$), fear of infection ($n = 9$), stigmatisation ($n = 5$), and financial repercussions ($n = 5$). Seekers also encountered notable challenges ($n = 13$). Future preparedness necessitates improved healthcare infrastructure, resource optimisation, and comprehensive protocols. Lessons should inform strategies to mitigate disruptions and prioritise the well-being of providers and seekers in future outbreaks.

**Keywords:** COVID-19; health system; health services; outpatient department (OPD); elective health services; emergency health services; personal protective equipment (PPE); health care providers





## 1. Introduction

The World Health Organisation (WHO) declared coronavirus disease 2019 (COVID-19) a public health emergency of international concern on 30 January 2020, and after that declared it a pandemic on 11 March 2020 [1].

This outbreak has placed unprecedented demands on the health systems, health workforce, and communities worldwide [2–8]. India's story is no different [9,10]. Given the large population of over 1.3 billion, the government of India declared a total lockdown across the country as a part of its efforts to control the disease spread [11].

The COVID-19 outbreak has placed severe strain on the health services in India at all levels in hospitals, clinics, and health care centres, with large and rapid increases in demand for patient care [10,12,13]. Caring for COVID-19 patients whilst maintaining treatment for patients with other conditions presented a complex planning challenge [14–16]. Ensuring safe and timely care to both COVID-19 patients and those with other conditions was a crucial aspect of the health system's response to this crisis. In order to free up enough

capacity to deal with the initial peak of the pandemic, the health system was forced to shut down or significantly reduce many areas of non-COVID-19 care across healthcare facilities during April, May, and June 2020 [17–20]. This, combined with fewer patients seeking care during the lockdown, meant that there had been a significant drop in elective procedures, urgent referrals, first treatments, and outpatient services [21,22]. A plethora of activities, adaptations, and measures were undertaken and implemented in a myriad of ways to protect lives and avoid a health system collapse [13,22–24]. Furthermore, the pandemic had profound mental and physical health implications for healthcare workers [25] and healthcare seekers, including individuals undergoing quarantine, hospitalisation, those who had recovered from the infection, and those managing other health conditions [15,17].

While several primary studies [26–28] have examined the impact of COVID-19 on the health system and its stakeholders in the Indian context, there is a need for a systematic appraisal and critique of these studies, particularly focusing on the peak of the outbreak. This review aims to consolidate the existing research to gain a comprehensive understanding of the health system's response and assess the impact of the pandemic on healthcare services, healthcare providers, and healthcare seekers across various clinical areas and facilities during the peak of the COVID-19 outbreak in India. The review aims to achieve the following objectives in the Indian context:

A. Assess the impact of COVID-19 on the provision, utilisation, and availability of health services;
B. Understand the health system's response—adaptations, interventions and efforts for continuity and resumption of services;
C. Evaluate the implications of COVID-19 and its response on individuals—healthcare providers and healthcare seekers.

## 2. Materials and Methods

The review is registered with PROSPERO (International Prospective Register of systematic reviews) at the National Institute for Health Research and Centre for Reviews and Dissemination (CRD) at the University of York, UK (registration number CRD42020227327) (Link in Supplementary Materials). Design and reporting were conducted per the PRISMA Statement [29,30].

### 2.1. Inclusion Criteria

- Studies conducted exclusively within the Indian context, from the onset of COVID-19 to its first peak and in its aftermath until 15 December 2020;
- Primary studies of any research design;
- Studies evaluating the impact of COVID-19 on the provision, utilisation, and availability of health services;
- Studies exploring the health system's response in terms of adaptations, interventions and efforts made in different types of health facilities for maintaining the continuity of services;
- Studies examining the impact of the pandemic on individuals—health care providers, individuals with acute and chronic diseases.

### 2.2. Exclusion Criteria

- Studies on pandemics other than COVID-19;
- Studies that are not primary (such as reviews, reports, policy briefs, commentary etc.);
- Global/multi-country studies with India just as one of the settings and studies not conducted in the Indian context;
- Studies evaluating aetiology, pathophysiology, histopathology, serology or laboratory examination of COVID-19, clinical trials, or vaccine development;
- Studies that are not written in English.

*2.3. Search Strategy*

We utilised the Preferred Reporting Items for Systematic Reviews and Meta-Analyses (PRISMA) guidelines to conduct a search of electronic databases. We electronically searched six databases, namely PubMed, MEDLINE, Embase, Global Health, CINAHL, and the WHO database on COVID-19, to identify primary studies employing quantitative, qualitative, and mixed methods. The search was limited to primary studies published in English until 15 December 2020. Our search approach employed a combination of Medical Subject Headings (MeSH), free-text terms, and word variants related to COVID-19, the health system, health services, communities, and community facilitators in the Indian context. In addition to the electronic databases, we also conducted a search on Google Scholar for relevant primary studies. Furthermore, we performed a snowball search to identify references from relevant papers. Detailed search strategies were developed for each electronic database. An illustration of the search strategy and search terms is given in Figure 1:

**Search strategy and terms in PubMed**

| 1. | Search: COVID-19 or coronavirus or severe acute respiratory syndrome coronavirus 2 or ncov or 2019-nCoV or COVID-19 or SARS-CoV-2 | 99,129 |
|---|---|---|
| 2. | Health system or Delivery of Health Care or Health Information Systems or Delivery of Health Care, Integrated or Community Health Planning or Sentinel Surveillance | 1,829,467 |
| 3. | Health service* or Reproductive Health Service* or Urban Health Service* or Suburban Health Service* or Adolescent Health Service* or Women's Health Service* or Preventive Health Service* or Mental Health Service* or Maternal Health Service* or non communicable disease treatment | 972,185 |
| 4. | Health care provider* or Health Personnel* or Mandatory Testing or Insurance, Health, Reimbursement or Alert Fatigue, Health Personnel or Mass Casualty Incident* | 425,046 |
| 5. | Community facilitator* or community engagement or community leader* | 66,394 |
| 6. | Community* or Population or household* or Individual* or People | 12,460,095 |
| 7. | India or Indian state* or Indian population | 633,358 |
| 8. | 2 OR 3 OR 4 OR 5 OR 6 | 13,296,398 |
| 9. | 1 AND 7 AND 8 | 2606 |
| 10. | 9 AND Filters: Full text, Evaluation Study, Journal Article, Multicentre Study, Observational Study, Randomized Controlled Trial, Validation Study, English, Humans | 922 |

**Search strategy in Google Scholar—screen the first 10 pages of results Sorted by relevance:**

(COVID-19 OR coronavirus OR severe acute respiratory syndrome coronavirus 2 OR ncov OR 2019-nCoV OR COVID-19 OR SARS-CoV-2) AND (Health system OR Delivery of Health Care OR Health Information Systems OR Delivery of Health Care, Integrated OR Community Health Planning OR Sentinel Surveillance) AND(Health service* OR Reproductive Health Service* OR Urban Health Service* OR Suburban Health Service* OR Adolescent Health Service* OR Women's Health Service* OR Preventive Health Service* OR Mental Health Service* OR Maternal Health Service* OR Non communicable disease treatment) AND(Health care provider* OR Health Personnel* OR Mandatory Testing OR Insurance, Health, Reimbursement OR Alert Fatigue, Health Personnel OR Mass Casualty Incidents) AND(Community facilitator* OR community engagement OR community leader*) AND(Communit* OR Population OR household* OR Individual* OR People) AND (India OR Indian state* OR Indian population)

**Figure 1.** Search strategy and terms in databases (PubMed and others) and Google Scholar.

*2.4. Data Extraction (Selection and Coding)*

The literature search results were meticulously organised and stored using the Mendeley reference management software. By importing the results into this software, we were able to seamlessly access full-text articles, annotate important information, remove duplicates, and effortlessly generate citations and a bibliography in the desired format [31–35]. Mendeley played a pivotal role in streamlining these tasks and optimising the management of the literature search results.

To maintain data integrity, we utilised the duplicate identification feature in Mendeley to identify and eliminate any redundant records. This step ensured that we worked with a unique set of unduplicated references. Subsequently, these unique records were imported into Covidence, a web-based systematic review software package developed by Veritas Health Innovation [36]. The two-stage screening process was then conducted in Covidence, facilitating collaboration among authors by providing an adequate platform to discuss and make study selections [37–39].

In the first stage of screening, the two authors independently assessed the titles and abstracts of the retrieved records against the predefined inclusion and exclusion criteria. This process helped us narrow down the selection to studies that potentially met the criteria.

Subsequently, all these studies selected in the initial screening stage underwent a thorough full-text screening process conducted independently by the two authors. During this second stage, the study design, relevance of the outcome measures and findings to the specified objectives of the review, and the targeted population were meticulously investigated. Based on this, thorough evaluation, final decisions were made on the inclusion of each study in the review. Studies that did not meet the inclusion criteria after the two-stage screening process were dropped from further consideration.

Any disagreements or uncertainties that arose during the review process were carefully addressed through discussions between the authors. In cases where specific issues required additional expertise, such as determining the exclusion of commentaries or perspectives, or assessing the relevance of outcome measures in the study to the objectives of the review, an independent reviewer was engaged to provide valuable insights and a fresh perspective. Covidence served as a crucial platform for documenting screening decisions, resolving conflicts, and ensuring transparency in our decision making. Its user-friendly interface and collaborative features facilitated effective communication and consensus-building among the authors.

Once the final set of included studies was determined using Covidence, we exported the relevant information required from the studies, including study characteristics and outcomes, into a structured and pre-developed data extraction form within an Excel spreadsheet [38,40–42]. This form consisted of columns representing the variables of interest. Two reviewers independently extracted the necessary data from each study and entered them into the corresponding cells of the spreadsheet. Data extraction form in Excel aided in the systematic and structured extraction and management of data from multiple studies, leading to a comprehensive and well-organised review.

*2.5. Data Items*

We meticulously extracted and structured the following information from the included articles, aligning it with the review's stated objectives:

1. Author name(s);
2. Publishing journal;
3. Study design;
4. Location of the study;
5. Targeted population;
6. Sample size;
7. Outcome measures and findings in the included studies related to the current review objectives:

- Availability, provision, and delivery of health services mentioned in the studies: We extracted data on how the included studies discussed the availability and delivery of health services during the COVID-19 pandemic, addressing the first objective of our review.
- Adaptations and changes in the health system, along with efforts undertaken: We organised information about the adaptations and changes made within the health system to cope with the pandemic and any efforts undertaken to ensure continued healthcare delivery in relation to the second objective of our review.
- Impact on healthcare providers: Data related to the impact of COVID-19 on healthcare providers, including their working conditions, well-being, and any challenges faced, were collected, addressing the third objective of our review that takes into account healthcare providers.
- Impact on the health, livelihood, and disease progression among individuals and communities: We documented the impact of COVID-19 on the health and well-being of individuals and communities, including effects on livelihood and disease progression, pertaining to the third objective of our review related to healthcare seekers.

8. Limitations and recommendations

By organising the data in this manner, we could effectively analyse how the findings from the included articles related to the stated objectives, providing a comprehensive understanding of the impact of COVID-19 on the health system, healthcare providers, and individuals.

### 2.6. Outcomes

Considering the anticipated heterogeneity in the widespread impact of COVID-19 on the health system and its various components and stakeholders, attributed to a wide range of settings, medical conditions and specialities, types of health system responses, changes and adaptations, healthcare providers' experiences, and healthcare seekers' conditions, our review included studies reporting a broad range of outcomes, addressing the study objectives. Our primary focus was on studies examining the effects on at least one primary or secondary outcomes, listed below:

2.6.1. Primary

1. Health services for varied health conditions:
   - Changes in the provision, utilisation, or availability of:
     ○ Outpatient department (OPD);
     ○ Elective health services;
     ○ Emergency health services

     pertaining to reproductive, maternal, and child health services, non-communicable diseases (cancer, cardiovascular diseases, diabetes, etc.), and other general or specific health conditions (orthopaedic, ophthalmic, neurological, etc.).

2. Health system response to COVID-19:
   - Efforts and adaptations made in:
     ○ General precautionary and infection prevention measures;
     ○ Protocols and guidelines;
     ○ Staff allocation, management, and training;
     ○ Personal Protective Equipment (PPE);
     ○ Physical infrastructure and resources.

These outcomes address objectives A and B of our review, which aim to assess the impact of COVID-19 on the provision, utilisation, and availability of health services in the Indian context and understand the health system's response, including adaptations, interventions, and efforts for continuity and resumption of services.

2.6.2. Secondary

1. Mental/psychological health of health care providers:

   - Depression, anxiety, and burnout;
   - Fear of infecting themselves and transmitting it to family members;
   - Financial repercussions.

2. General health or disease conditions of health care seekers/individuals:

These outcomes pertain to objective C, focusing on evaluating the impact of COVID-19 and the health system's response on individuals, encompassing both healthcare providers and healthcare seekers.

*2.7. Critical Appraisal—Quality Assessment*

Two reviewers (A.S.C., A.N.) appraised the included studies using two tools:

1. CASP (Critical appraisal skills programme) checklist for qualitative (observational) studies [43–45]. This tool is commonly used for evaluating the quality and methodological rigour of qualitative studies. It helped us assess the appropriateness of the study design, data collection methods, data analysis, and the credibility of study findings in qualitative research studies included in the review.

2. AXIS critical appraisal tool for cross sectional studies/surveys [46–51]. This tool is designed to assess the quality of cross-sectional studies or surveys. It helped us evaluate such studies included in the review on various aspects, including sampling methods, participant selection, data collection, and statistical analysis.

Both tools have been adapted for use in this review. The reviewers utilised the checklists to evaluate each included study's design, conduct, and analysis, with a focus on addressing potential biases.

Any discrepancies in ratings on different questions/items between the two reviewers were thoroughly discussed, and consensus was reached for each checklist item. The quality appraisal was not a criterion for study inclusion; therefore, no studies were excluded based on the results of the appraisal.

*2.8. Data Analysis*

Due to the heterogeneity of the studies, a detailed narrative synthesis was employed to analyse and interpret the data based on the outcome measures. Narrative synthesis [52,53] refers to an approach to the systematic review and synthesis of findings from multiple studies that relies primarily on the use of words and text to summarise and explain the findings of the synthesis. It adopts a textual approach to the process of synthesis to "tell the story" of the findings from the included studies. This narrative synthesis approach enabled us to thoroughly examine and comprehend the data, taking into account its diverse nature. It played a crucial role in identifying patterns and trends within the findings, allowing us to gain valuable insights from the varied studies without compromising their heterogeneity.

**3. Results**

*3.1. Screening and Inclusion of Studies*

From the screening of five databases—PubMed, MEDLINE, Global Health, EMBASE, CINAHL, and WHO COVID-19 Global literature on coronavirus disease—we identified 1477 records (Figure 2). Additionally, 74 records were retrieved and added from other sources, such as Google Scholar and references in the citations. After eliminating duplicates, we retained 1271 unique records. The duplication mainly occurred because certain articles were indexed in multiple databases. Subsequently, 526 studies were selected based on the initial screening of abstracts. The full text of these 526 articles was assessed in detail for adherence to eligibility criteria, resulting in 121 articles. Of these, 38 articles were included for narrative analysis, while 83 were excluded for not meeting the eligibility criteria.

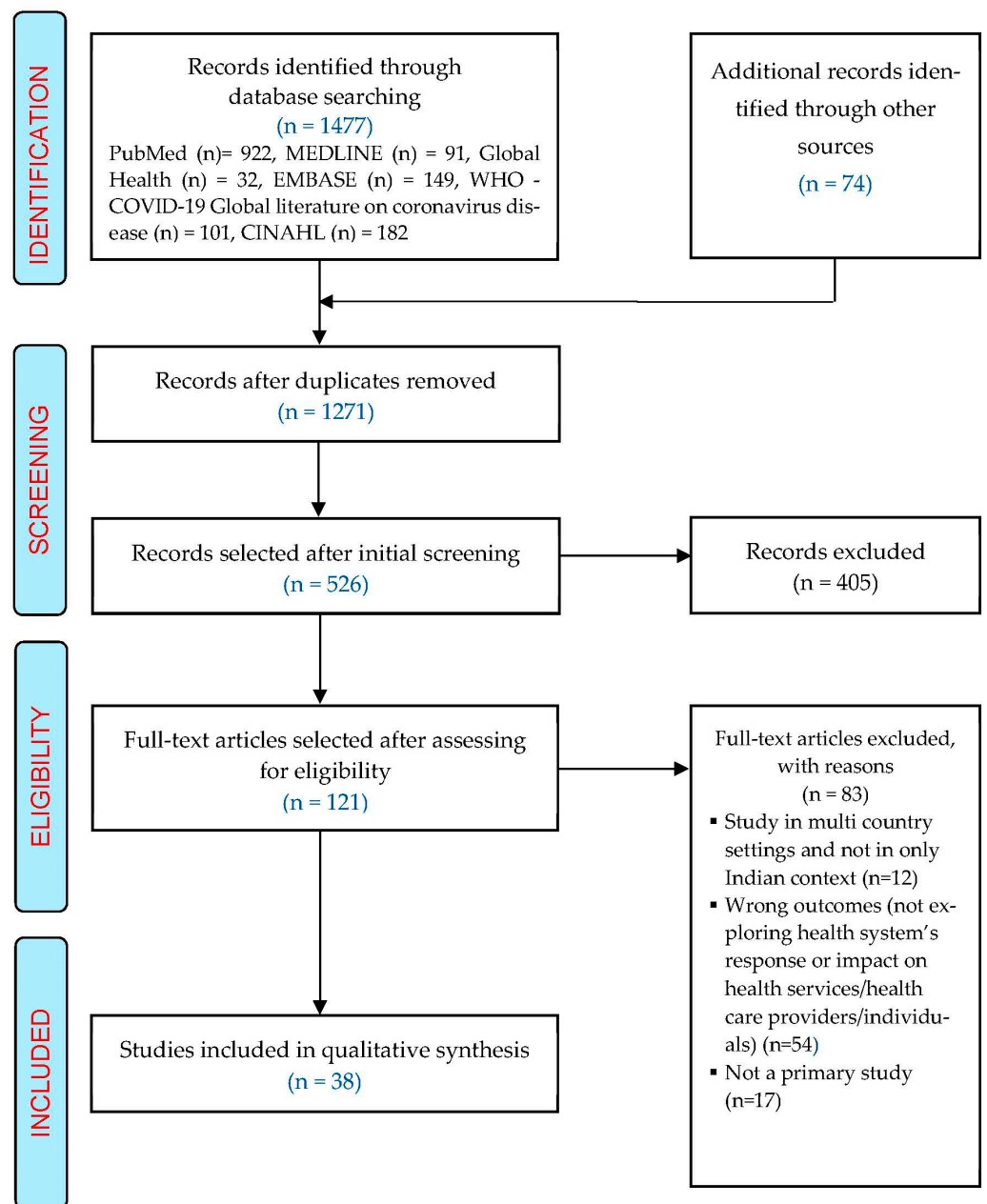

**Figure 2.** PRISMA Flow diagram.

### 3.2. Narrative Analysis

The review comprises a total of 38 primary studies (Table 1) with a collective participant count of 22,502. These studies consist of 21 cross-sectional surveys, 16 observational studies, and one mixed-method study combining survey and observational approaches. They were conducted in diverse settings, including clinics, hospitals, and primary healthcare or community health centres, which might have affected the choice of the study design, targeted population, outcome measures, and, eventually, the findings in these studies.

**Table 1.** Characteristics of Included Studies in Review—Authors, Areas of Work, Participant Types and Numbers, Settings/Contexts, Outcome Measures, and Findings from 38 Studies (in relation to review's objectives).

| S. No. | Author | Area of Work | Participants | No. of Participants | Setting/Context | Main Outcome Measures and Findings from the Included Studies Aligned with the Review's Objectives |
|---|---|---|---|---|---|---|
| 1. | Gautam P et al. [54] | Cancer care | Onco-surgeons | 15 | A tertiary care centre, Pune | *Availability, provision, and delivery of health services*<br>❖ OPDs operated by appointment only to prevent overcrowding.<br>❖ Deferral of elective procedures; some patients sent for neoadjuvant treatment.<br>❖ Minimal Invasive Surgeries (MIS) practiced by a few surgeons with successful use of smoke evacuators.<br>❖ Over 50% doubted COVID-19 aerosol spread during MIS.<br>❖ 53.8% believed cancers would be upstaged due to lockdown.<br>❖ 46.2% expected higher mortality rates in cancer patients.<br><br>*Adaptations and changes*<br>**General precautionary and infection prevention measures:**<br>❖ Entry screening, regulated attendance, hand sanitizers, and hygiene promotion.<br>Ő Marked standing spaces, spaced sitting areas, and regular surface sanitization.<br><br>**Guidelines and Protocols**<br>❖ A group of management personnel, surgeons, physicians, intensivists, and nursing supervisors formulated the standard operating procedures (general hospital, staff, surgical oncology, preoperative, OT and post operative protocols and guidelines) based on the scanty literature available.<br>❖ Continued services and operations with strict adherence to protocols.<br><br>**Staff management, allocation, and training:**<br>❖ Staff divided into COVID-19 and non-COVID-19 teams, with isolated ward staff adequately quarantined.<br>❖ All staff trained in safety precautions, including PPE usage<br><br>**Preoperative/OT/post-operative measures:**<br>❖ All preoperative investigations on the first visit.<br>❖ COVID-19 swab test for every pre-op patient, allowing only negative patients for surgery.<br>❖ High-risk elderly and frail patients requested to wait or placed on neoadjuvant therapy.<br><br>**PPE:**<br>❖ Increased PPE usage in most surgeons during operations.<br><br>**Infrastructure and resources:**<br>❖ Separate building allocated for managing COVID-19 patients, with about 170 beds, including an ICU.<br>❖ Sitting areas were adequately spaced to maintain social distancing<br>❖ The number of persons entering crowded places (such as elevators) was limited and standing spaces marked.<br>❖ Air exchange was maintained OTs with ultra-filtration for ventilation |

**Table 1.** *Cont.*

| S. No. | Author | Area of Work | Participants | No. of Participants | Setting/Context | Main Outcome Measures and Findings from the Included Studies Aligned with the Review's Objectives |
|---|---|---|---|---|---|---|
| | | | | | | *Impact on Healthcare providers*<br><br>❖ Fear of contracting COVID-19 increased<br>❖ Majority experienced income decline, 53.8% considered financial changes.<br>❖ Mental stress of not being able to operate, staying at home financial loss, or some other reason<br>❖ Surgeons' eagerness to work remained despite colleague's COVID-19 case.<br>❖ Irritation level with relatives' calls showed little change.<br>❖ Nearly all continued working with safety precautions. |
| 2. | Goyal N et al. [55] | Neurosurgery | Patients operated under the Department of Neurosurgery since the onset of pandemic | 164 | The Department of Neurosurgery, All India Institute of Medical Sciences, Uttarakhand institute | *Availability, provision, and delivery of health services*<br><br>❖ Total surgeries decreased by 52.2% compared to the previous year.<br>❖ Cancellation of elective surgery cases.<br>❖ Non-emergent cases decreased from 57.7% to 11.3%.<br><br>*Adaptations and changes*<br>**Staff management, allocation, and training:**<br><br>❖ Out-station leaves of all HCWs cancelled to preserve workforce availability.<br>❖ Decreased staffing to prevent fatigue.<br>❖ Remote work for clinical, resident, and support staff.<br>❖ Staff re-deployment to COVID-19 areas.<br>❖ Physical meetings replaced by virtual meetings.<br>❖ Restricted visitors to facilities.<br>❖ Virtual COVID-19 educational programs for HCWs.<br><br>**PPE:**<br><br>❖ On-site employees required to wear face masks daily.<br>❖ A level-III PPE) is worn by all in the operation room (OR)<br><br>**Preoperative/OT/post-operative measures:**<br><br>❖ Screening OPD for COVID-19 symptoms and travel history.<br>❖ Thermal screening and COVID-19 testing.<br><br>**Infrastructure and resources:**<br><br>❖ Telemedicine to conduct non-emergency outpatient visits |

| S. No. | Author | Area of Work | Participants | No. of Participants | Setting/Context | Main Outcome Measures and Findings from the Included Studies Aligned with the Review's Objectives |
|---|---|---|---|---|---|---|
| 3. | Nair A et al. [56] | Ophthalmic practice | Ophthalmologists | 1260 | Online survey of ophthalmologists from Private clinics, ophthalmic institutes, corporate/multi-specialty hospitals, government/municipal hospitals, freelancing surgeons | *Availability, provision, and delivery of health services*<br>❖ Majority (72.5%) stopped all clinical work during the lockdown.<br>❖ Trauma cases (51.9%) were the most common, and 82.9% were seeing only emergency cases.<br>❖ 81.8% performed only emergency surgical procedures.<br>❖ 11.8% would immediately resume elective surgeries after easing restrictions.<br>❖ 30.4% planned to wait for at least one week.<br>❖ 57.8% were unsure about when to start elective surgeries after resuming clinical duties.<br><br>*Adaptations and changes*<br>**PPE**<br>❖ 16.6% preferred additional PPE for surgeries.<br>**Preoperative/OT/post-operative measures:**<br>❖ 9.9% preferred pre-operative COVID-19 testing for elective surgeries.<br>❖ 10.7% favoured pre-operative screening and PPE use for all surgeries.<br><br>*Impact on Healthcare providers*<br>❖ Fear and stress: 59% of ophthalmologists believed they faced a higher COVID-19 risk than other specialties during patient exams.<br>❖ 77.5% of stand-alone private practitioners closed their practices due to lack of in-patient facilities for isolation.<br><br>**Infrastructure and resources:**<br>❖ 77.5% had begun telephonic/e-mail/video consultations or consultations over social media applications |
| 4. | Singh M et al. [57] | Palliative Cancer care | Patients availing palliative care service | 1161 | Department of pain and palliative medicine in a tertiary comprehensive Cancer centre, Gujarat. | *Availability, provision, and delivery of health services*<br>❖ Patients were treated by the pain and palliative care team, with most suffering from head-and-neck cancer.<br><br>*Adaptations and changes*<br>**Precautionary measures:**<br>❖ Precautions were taken during OPD and emergency consultations to avoid virus transmission.<br>❖ Entry inside the OPD with masks only, social distancing, and limited patients in the room at a time.<br>❖ Regular handwashing and sanitizer use.<br><br>**Staff management, allocation, and training:**<br>❖ Mandatory training for infection control, sanitization, and COVID-19 patient management for all staff and doctors.<br>❖ Emphasis on staff well-being through active listening, support, and monitoring. |

**Table 1.** *Cont.*

| S. No. | Author | Area of Work | Participants | No. of Participants | Setting/Context | Main Outcome Measures and Findings from the Included Studies Aligned with the Review's Objectives |
|---|---|---|---|---|---|---|
| | | | | | | **PPE:**<br>❖ Proper universal precautions using PPE kits for treating patients with severe symptoms visiting emergency.<br>**Infrastructure and resources:**<br>❖ Telemedicine consultations were encouraged for patients.<br>*Impact on Healthcare Seekers*<br>❖ Fear and increased Risk: Patients travelled far from other states (e.g., Karnataka and Odisha) to access medical care, risking their immunocompromised state. The process was financially draining and challenging for both patients and accompanying persons.<br>❖ Medicine unavailability: Due to regulations, opioids' availability is limited in India, prompting patients to travel long distances to find them as nearby centres often do not have adequate supply. |
| 5. | Garg S et al. [58] | Primary health care | Supervisors and managers of primary health care facilities | 51 | Primary health care facilities attached to medical colleges and institutions anywhere in India, either in the government or private setting. | *Availability, provision, and delivery of health services*<br>❖ Before COVID-19, participant institutions provided on average antenatal care (ANC) to 26.5 clients and immunisation services to 41.4 clients.<br>❖ Outpatient services were significantly disrupted during the COVID-19 epidemic, with NCD and immunization clinics experiencing the most reduction, while ANC and general OPDs were less affected.<br>❖ Fever (flu) clinics were implemented at 72.4% ($n = 37$) of the sites to screen patients for suspected COVID-19 and provide appropriate referral services when needed.<br>*Adaptations and changes*<br>**Precautionary measures:**<br>❖ On average, each site had a patient queuing capacity of 14.1 persons with minimum physical distancing.<br>❖ Almost half of the sites were missing separate or multiple entries and exits, and a majority reported inadequate ventilation.<br>❖ Airborne infection control measures were absent in most sites, and adequate handwashing services for patients were unavailable at 23.5% of sites.<br>❖ Staff management, allocation, and training:<br>❖ 78.4% of the sites had previously provided training on managing patients with presumptive COVID-19 to their health staff.<br>**PPE:**<br>❖ PPE suits were available at 27.4% of sites, N95 masks at 50.9% of sites, and surgical masks at 39.3% of sites.<br>❖ Only half of the medical colleges and institutions provided N95 masks to healthcare providers at their primary health care facilities. |

| S. No. | Author | Area of Work | Participants | No. of Participants | Setting/Context | Main Outcome Measures and Findings from the Included Studies Aligned with the Review's Objectives |
|---|---|---|---|---|---|---|
| | | | | | | *Physical infrastructure and resources*<br>❖ Each site, on average, reported a patient queuing capacity of 14.1 persons to maintain social distancing.<br>❖ Nearly half (49%) of the sites were missing separate or multiple entry and exits (52.9%)<br>❖ A majority (57%) of the participants reported inadequate ventilation at their PHC sites.<br>❖ Fever (flu) clinics had been started at 72% of the sites to screen patients and initiate appropriate referral services.<br>❖ Impact on Healthcare Providers<br>❖ Lack of Confidence in COVID-19 Patient Segregation<br>❖ Hesitancy in Operating Dedicated Fever Clinics |
| 6. | Nasta AM et al. [59] | General Surgical practice | Surgeons (members of Indian Association of Gastro-intestinal Endo-surgeons -IAGES) | 153 | Online survey of members of Indian Association of Gastro-intestinal Endo-surgeons (IAGES) | *Availability, provision, and delivery of health services*<br>❖ 36.5% completely stopped outpatient services, and 63.5% of surgeons had a reduction in their services.<br>❖ 93.3% stopped all elective surgical work, while 5.2% scaled down elective surgeries.<br>❖ 77% performed no elective procedures, and 16% performed less than 5 surgeries during the lockdown period.<br>❖ 9% of surgeons performed no emergency surgeries, and 42.5% reduced emergency services where feasible.<br>❖ Average elective and emergency surgeries performed in the month of lockdown were merely one and five, respectively.<br><br>*Adaptations and changes*<br>**PPE:**<br>❖ 56.9% of surgeons said they would use PPE in all cases for safe surgical practice.<br>**Protocol and Guidelines:**<br>❖ 71.5% stated there are insufficient guidelines for future surgical practice in terms of safety.<br>❖ There is a definite need for guidelines regarding safety for future surgical practices and solutions to overcome financial liabilities in the near future.<br>**Preoperative/OT/post-operative measures:**<br>❖ Hydroxychloroquine (HCQ) was taken by 52% surgeons for chemoprophylaxis.<br>❖ 38% preferred open surgery, and 33% preferred using filters for de-sufflation.<br>❖ 93% surgeons felt that laparoscopic surgery and use of energy sources increased the risk of aerosol spread of the virus.<br>**Infrastructure and resources**<br>❖ 50% started online consultations. |

| S. No. | Author | Area of Work | Participants | No. of Participants | Setting/Context | Main Outcome Measures and Findings from the Included Studies Aligned with the Review's Objectives |
|---|---|---|---|---|---|---|
| | | | | | | *Impact on Healthcare Providers*<br>**Financial impact**<br>❖ Income reduction: Surgeons and healthcare providers faced significant drop in monthly income.<br>❖ Variation by hospital type: Private hospital surgeons experienced greater income reduction compared to government counterparts.<br>❖ Hospital ownership: 33% of respondents owned hospitals with a monthly financial liability of 2.25 million rupees (30,000 USD). |
| 7. | Choudhary R et al. [60] | Cardiovascular emergencies | Patients presenting with cardiovascular emergencies | 289 | Four tertiary regional emergency departments in western India | *Availability, provision, and delivery of health services*<br>Admissions decrease and urgency<br>❖ 67% reduction in STEMI admissions during the lockdown compared to the pre-COVID-19 period.<br>❖ 93% reduction in NSTEMI admissions during the same interval.<br>❖ Increase in the proportion of ACS patients with delayed presentations and mechanical complications (myocardial dysfunction, heart failure, and cardiogenic shock).<br>❖ Alarming decrease in urgent hospitalisations for Acute Decompensated Heart Failure (ADHF) and high-degree/complete AV block, despite their critical diagnostic and management requirements. |
| 8. | Goyal M et al. [61] | Maternal health | Pregnant women | 633 | Department of Obstetrics and Gynaecology at All India Institute of Medical Sciences, Jodhpur | *Availability, provision, and delivery of health services*<br>❖ Decrease in admissions: About 43.27% decrease during the pandemic.<br>❖ Decline in deliveries: Significant 45.1% decrease in deliveries compared to pre-COVID-19 times (p < 0.001).<br>❖ Increase in ICU admissions: More patients required ICU admission during the pandemic compared to pre-COVID-19 period (p < 0.05).<br>❖ Insufficient antenatal visits: About 32.5% of pregnant women had fewer antenatal visits than recommended. Many women avoided routine check-ups during the strict lockdown, lasting at least 3 months<br><br>*Adaptations and changes*<br>**PPE:**<br>❖ Expedited PPE production and widespread use<br>**Preoperative/OT/post-operative measures:**<br>❖ COVID-19 tests were conducted before admission and suspected cases were admitted in isolation units.<br>❖ Those who tested positive for COVID-19 were managed at the centre with good maternal and foetal outcome.<br>❖ Emergency services were continued, and all patients were treated, whether they were registered or not. |

| S. No. | Author | Area of Work | Participants | No. of Participants | Setting/Context | Main Outcome Measures and Findings from the Included Studies Aligned with the Review's Objectives |
|--------|--------|--------------|--------------|---------------------|-----------------|------------------------------------------------------------------------------------------------------|
|        |        |              |              |                     |                 | **Infrastructure and resources**<br><br>❖ Provided antenatal services through telemedicine,<br>❖ Impact on Healthcare Seekers<br>❖ Surge in high-risk pregnancies and complications aggravated by delayed care.<br>❖ Reasons for delay: strict lockdown and lack of transportation, fear of infection.<br>❖ Neglected conditions: anaemia, pregnancy-induced hypertension.<br>❖ Preference for home-based care.<br>❖ Late presentations: post-dated pregnancies, advanced labour.<br>❖ Serious complications: eclampsia, acute renal failure, sepsis, pneumonia. |
| 9. | Deshmukh, S [62] | Cancer | Cancer patients who underwent surgery, CT, and RT | 553 | A charitable cancer hospital, Pune | *Availability, provision, and delivery of health services*<br><br>❖ Radiotherapy patients' treatment decreased by nearly 40% during the first month of lockdown, while surgery experienced an 80% decrease.<br>❖ Number of CT patients also dropped significantly but recovered in the 4th week with the implementation of SOP and safe working protocols.<br>❖ Most CT patients received single-drug therapy to reduce hospital admissions and visits.<br>❖ Decrease in surgeries due to deferral and use of neoadjuvant treatment, as well as apprehension about surgeries in the situation.<br><br>*Adaptations and Changes*<br>**Protocol and Guidelines:**<br><br>❖ Patient appointments for radiation therapy maintained to ensure consistency in treatment teams.<br>❖ Strict protocols for intraoperative, postoperative, and emergency cases implemented in each department.<br><br>**Staff management, allocation, and training:**<br><br>❖ Staff divided into two groups with different working times to prevent overlap.<br>❖ Quarantine and testing required if any staff or patient tests positive for COVID-19.<br><br>**PPE:**<br><br>❖ Universal PPE usage by all HCWs who come in direct contact with patients or infected waste.<br>❖ Surgical team with extra face and eye protection, maintaining sterility<br><br>**Preoperative/OT/post-operative measures:**<br>**Preoperative:**<br><br>❖ Surgery planned 8 to 10 days after consultation.<br>❖ Strict home quarantine advised before admission.<br>❖ COVID-19 testing conducted.<br>❖ Asymptomatic patients observed for 72 h (5 days for some).<br>❖ Patients counselled about risks and informed consent obtained. |

**Table 1.** *Cont.*

| S. No. | Author | Area of Work | Participants | No. of Participants | Setting/Context | Main Outcome Measures and Findings from the Included Studies Aligned with the Review's Objectives |
|---|---|---|---|---|---|---|
| 9. | Deshmukh, S [62] | Cancer | Cancer patients who underwent surgery, CT, and RT | 553 | A charitable cancer hospital, Pune | **Intraoperative**<br>❖ Spaced surgeries for sterilisation.<br>❖ Rapid Sequence Induction for intubation.<br>❖ Smooth extubating under a plastic drape.<br>❖ Few patients with weekly and biweekly dose dense regimens of chemotherapy changed to 3 weekly protocols<br><br>**Infrastructure and resources:**<br>❖ Isolated ICU for Life threatening emergencies and asymptomatic patients were shifted to the main ICU after negative COVID-19 status.<br>❖ Designated area for Donning and doffing of PPE.<br>❖ Social distancing was maintained between patient beds (distance of 6 feet).<br>❖ The number of patients admitted for CT was limited to maintain proper social distancing measures.<br>❖ Air conditioner maintenance and filter cleaning were carried out to maintain proper ventilation.<br>❖ Telephonic consultations were encouraged for minor ailments.<br><br>*Impact on Healthcare Seekers*<br>❖ Fear of COVID-19 infections was the primary reason for the decrease in patient visits.<br>❖ Television and online news coverage heightened patient fear and anxiety.<br>❖ Unavailability of public transport and strict lockdown restrictions made private transport difficult.<br>❖ With few fuelling stations open, it became harder for patients to reach hospitals.<br>❖ Significant cost escalation deterred patients from undergoing planned medical procedures.<br><br>*Impact on Healthcare Providers*<br>❖ HCPs experienced significant behavioural changes, stress and fear.<br>❖ Operation theatre health workers socially ostracised<br>❖ The fear of infection and social isolation had a profound effect on the well-being and social interactions of HCWs. |
| 10. | Keshav K et al. [63] | Orthopaedic Practice | Practicing orthopaedic surgeons | 533 | Online nationwide survey | *Availability, provision, and delivery of health services*<br>❖ Over 90% of practicing surgeons experienced a significant drop in outpatient visits.<br>❖ 64% of surgeons stopped elective surgeries completely, and 21% stopped emergency surgeries as well.<br><br>**Infrastructure and resources:**<br>❖ 18% started telemedicine consultations |

**Table 1.** *Cont.*

| S. No. | Author | Area of Work | Participants | No. of Participants | Setting/Context | Main Outcome Measures and Findings from the Included Studies Aligned with the Review's Objectives |
|---|---|---|---|---|---|---|
| | | | | | | *Impact on Healthcare Providers*<br>❖ 28% of private sector doctors saw over 90% income reduction.<br>❖ 18.4% of government sector doctors had stable earnings.<br>❖ More experienced doctors (20–30+ years) were significantly affected ($p < 0.001$). |
| 11. | Gupta A et al. [64] | Head and neck cancer | Head and neck health care stakeholders | 16 | Major head and neck health care facilities across India | *Availability, provision, and delivery of health services*<br>❖ 50% of the hospitals were dedicated cancer centres, while the rest were tertiary care institutions with HNC facilities.<br>❖ 69% of the institutions halted outpatient facilities and elective surgeries, continuing only emergency admissions and surgeries.<br>❖ 31% of the institutes provided all types of HNC services, but with increased outpatient volume and decreased operations due to safety precautions and limited availability of PPE.<br><br>*Adaptations and Changes*<br>**Protocol and Guidelines:**<br>❖ 69% of the institutes followed institutional guidelines, while others adhered to state or other guidelines.<br>**PPE:**<br>❖ On-site employees required to wear face masks daily.<br>❖ Availability of PPE and N95 masks was limited in most centres (63%), leading to judicious use.<br>**Preoperative/OT/post-operative measures:**<br>❖ Preoperative COVID-19 testing was not allowed by government regulations, making decision-making for surgery based on COVID-19 status difficult.<br>**Infrastructure and resources:**<br>❖ A few (12.5%), institutions have started telemedicine consultations. |
| 12. | Sahu D et al. [65] | Orthopaedic surgery | Orthopaedic surgeons | 611 | Online nationwide survey | *Availability, provision, and delivery of health services*<br>❖ Orthopaedic surgeons attended to patients based on urgency, with 33% attending to urgent cases, 27% not attending OPD at all, and 26% handling acute trauma cases.<br>❖ Many orthopaedic surgeons performed unavoidable trauma surgeries (62%) or no surgeries (35%) during the period.<br>❖ The majority (74.4%) only looked after the orthopaedic part, while 16.4% were not actively caring for any patients, and only a few (0.7%) directly took care of COVID-19 patients.<br><br>*Adaptations and Changes*<br>**PPE:**<br>❖ Many orthopaedic surgeons used a normal medical/surgical mask (61.4%) or N95 mask (24%) while attending patients.<br>❖ Few of them used masks and disposable gowns (14%) or full PPE as prescribed. |

| S. No. | Author | Area of Work | Participants | No. of Participants | Setting/Context | Main Outcome Measures and Findings from the Included Studies Aligned with the Review's Objectives |
|---|---|---|---|---|---|---|
| | | | | | | **Staff management, allocation, and training:** <br> ❖ Several orthopaedic surgeons were actively involved in teaching (10%) and learning (27%) through webinars or other means. <br> ❖ Impact on Healthcare Providers <br> ❖ Stress levels: 23% experienced extreme stress, while 41% had mild stress. <br> ❖ Reversed work-life balance: Majority (58%) felt their work-life balance was completely reversed. |
| 13. | George CE et al. [66] | Healthcare services in a large slum | A healthcare team of doctors, nurses, paramedical and support staff. | 87 | Community Health Division, Bangalore Baptist Hospital, Bangalore | *Availability, provision, and delivery of health services* <br> ❖ Providing patient care in a poorly constructed, ill-ventilated clinic in a crowded slum caused strain on the healthcare team. <br> ❖ Patient triaging led to long queues and prolonged waiting times, which were further disrupted by rains. <br><br> *Adaptations and Changes* <br> **Staff management, allocation, and training:** <br> ❖ Human resource constraints became a burden on the team when vulnerable staff (older, pregnant) were relieved from risky duties. <br> ❖ Existing staff had to juggle their time between designing new interventions and providing care. <br><br> **PPE:** <br> ❖ Universal use of barrier precautions (masks, gloves, gowns and eyewear) <br> ❖ Reusable water-resistant PPE was designed and rolled out <br> ❖ Disinfecting PPEs in a space-constrained slum clinic was challenging. <br> ❖ Attending to patients fully clad in PPEs exhausted HCWs physically. <br> ❖ Tolerance limit nearly 4 h <br><br> **Infrastructure and resources:** <br> ❖ Patient care activities moved outdoors, but indoor areas lacked proper ventilation. <br> ❖ Patient registration, laboratory work, and dental procedures still conducted indoors with compromised ventilation. <br> ❖ Patient triaging resulted in long queues and prolonged waiting time. <br> ❖ Shortage of essential supplies (PPE, medicines) <br> ❖ Mounting operating costs due to increased measures (e.g., using two vehicles to reduce crowding) <br> ❖ Remote consultation opportunities such as teleconsultation. |

| S. No. | Author | Area of Work | Participants | No. of Participants | Setting/Context | Main Outcome Measures and Findings from the Included Studies Aligned with the Review's Objectives |
|---|---|---|---|---|---|---|
| | | | | | | *Impact on Healthcare Providers*<br>❖ Experience of fear: 75% reported experiencing fear at some point.<br>❖ Coping mechanisms: Hobbies (20.3%) and family time (39.1%) were cited as means of emotional regulation.<br>❖ Major Themes from QUAL Interviews causing stress included fear of death, guilt of disease transmission, anxiety about violence and stigma in slums, and exhaustion among healthcare professionals. |
| 14. | Agarwal D et al. [67] | Ophthalmology | Patients of vitreoretinal surgery | 86 | A government tertiary eye care hospital | *Procedures and Services*<br>❖ The number of vitreoretinal surgeries significantly decreased during the study period compared to pre-COVID-19 times.<br>❖ Trauma-related vitreoretinal surgery volume reduced, while trauma at home, especially in children, increased sharply during the lockdown.<br>❖ Immediate surgical intervention among children also decreased, possibly due to difficulty in accessing healthcare facilities.<br><br>*Adaptations and Changes*<br>**Precautionary Measures:**<br>❖ Patients were instructed on proper mask usage, social distancing, and hand sanitization at every point of contact.<br>❖ Healthcare workers used N95 masks, face-shields, disposable gowns, and gloves while providing patient care.<br>❖ Contact tracing was initiated for COVID-19 positive patients.<br><br>**Protocols and Guidelines:**<br>❖ A modified working protocol was adopted, focusing on enforcing COVID-19 precautions and forming dedicated infection control and disease surveillance committees.<br>❖ Modifications were made in the sterilisation protocol<br><br>**Staff management, allocation, and training:**<br>❖ The OT functioned with minimal staff in teams/shifts.<br>❖ Some vitreoretinal surgeries used a "heads-up" 3D visualisation system to minimise exposure risks.<br>❖ Only one attendant was allowed for each patient, and high-risk contacts were quarantined.<br><br>**Preoperative/OT/post-operative measures:**<br>❖ COVID-19 positive patients shifted to designated care facility.<br>❖ Isolation of the patient ensured.<br>❖ Using "heads-up" 3D visualisation system to increase the distance between the surgeon and the patient minimising the risk of exposure<br>❖ Use of passive polaroid glasses<br>❖ Use of povidone-iodine before surgery initiation<br>❖ Valved cannulas used, limited diathermy. |

| S. No. | Author | Area of Work | Participants | No. of Participants | Setting/Context | Main Outcome Measures and Findings from the Included Studies Aligned with the Review's Objectives |
|---|---|---|---|---|---|---|
| 15. | Barik S et al. [68] | Orthopaedic | Orthopaedic residents | 158 | Seven tertiary care centres in North India | *Availability, provision, and delivery of health services* <br> ❖ Working in the operating room (OR) and clinical examination of patients became difficult during the pandemic. <br> ❖ Working at dressing and plaster rooms, and sending laboratory or radiological investigations in the OPD were also challenging. <br><br> *Adaptations and Changes* <br> **Staff management, allocation, and training:** <br> ❖ Residents found learning through web platforms easier than before (44.2%). <br> ❖ However, participating in online case presentations and maintaining audience attention during online presentations were relatively difficult compared to offline activities. <br><br> **PPE:** <br> ❖ Obtaining PPE was challenging in the OR, OPD, and IPD settings. <br><br> *Impact on Healthcare Providers* <br> ❖ Majority of residents (51.3% in IPD, 53.2% in OR, and 56.3% in OPD) worked with anxiety about contracting COVID-19 infection. <br> ❖ Difficulty in spending time during quarantine alone (40.4%). <br> ❖ Challenges in pursuing non-orthopaedic hospital duties (59.5%) and socialising with others (48.9%). |
| 16. | Khurana DK et al. [69] | Palliative care | All patients coming to the clinic and inpatient referrals. | 108 | Pain and palliative care unit at a tertiary care hospital | *Availability, provision, and delivery of health services* <br> ❖ Despite the complete lockdown, the number of patients progressively increased, with a total of 108 patients visiting the clinic. <br> ❖ 78% of the patients were from Delhi, likely due to difficulties in crossing state borders under lockdown restrictions. <br> ❖ The main reason for visits was new-onset pain caused by noncompliance with drugs, as the opioid stock finished with the patient. <br><br> *Adaptations and changes* <br> **General Precautionary and Infection Prevention Measures:** <br> ❖ Regular personal and environmental sanitization was conducted. <br> ❖ Patients were checked for temperature using a non-contact thermometer, and hand sanitization and wearing masks were mandatory for patients and accompanying persons. <br> ❖ Social distancing was ensured by limiting the staff to one doctor and nurse per day. <br><br> **PPE:** <br> ❖ Staff in the OPD wore personal protective equipment (PPE) comprising N95 masks, glasses/face shields, and gloves. <br> ❖ When visiting patients in the ward, staff also wore a full PPE kit. |

**Table 1.** *Cont.*

| S. No. | Author | Area of Work | Participants | No. of Participants | Setting/Context | Main Outcome Measures and Findings from the Included Studies Aligned with the Review's Objectives |
|---|---|---|---|---|---|---|
| | | | | | | **Infrastructure and resources:**<br>❖ Telemedicine using phone and video calls was implemented and found to be useful.<br>*Impact on Healthcare seekers*<br>❖ Patient suffering and opioid requirement: Patients sought hospitals due to the agony of pain and suffering, as evidenced by the need for opioids. |
| 17. | Wilson W et al. [27] | COVID-19 | HCPs (doctors and nurses) directly involved in the triage, screening, diagnosing, and treatment of COVID-19 patients and suspects. | 433 | Online survey—Ten states and one union territory | *Adaptations and Changes*<br>**PPE**<br>❖ 50.0% expressed dissatisfaction with the availability of personal protective equipment (PPE).<br>*Impact on Healthcare Providers*<br>❖ High-level stress prevalence: 3.7%<br>❖ Depressive symptoms requiring treatment: Prevalence of 11.4%<br>❖ Anxiety symptoms requiring further evaluation: Prevalence of 17.7%<br>❖ Concerns about infection spread: 78.0% had serious concerns about spreading infection to friends or family. |
| 18. | Khasne RW et al. [70] | COVID-19Burnout/fear/stigma | All HCPs (doctors, nurses, paramedics) looking after COVID-19 patients. | 2026 | Nationwide online survey | *Availability, provision, and delivery of health services*<br>❖ 86% of respondents worked in high-risk areas.<br>❖ 98.8% believed that mental health was as important as physical health for healthcare workers (HCWs).<br>*Adaptations and Changes*<br>**Staff management, allocation, and training:**<br>❖ HCWs to receive frequent communications with information, instructions, training, and technical updates on COVID-19.<br>❖ A supportive working environment boost the confidence and morale of HCWs and aid in the recovery of those facing challenges.<br>❖ HCWs to focus on self-care and destressing by adopting personalised resilience plans and attending workshop-based training for psychological well-being.<br>**PPE:**<br>❖ Lack of Adequate PPE |

**Table 1.** *Cont.*

| S. No. | Author | Area of Work | Participants | No. of Participants | Setting/Context | Main Outcome Measures and Findings from the Included Studies Aligned with the Review's Objectives |
|---|---|---|---|---|---|---|
| | | | | | | *Impact on Healthcare Providers*<br>❖ Pandemic-related burnout score was significantly higher than personal and work-related burnout scores ($p < 0.01$).<br>❖ 55.3% of respondents feared contracting COVID-19 infection, while 66.9% feared carrying the infection home.<br>❖ 22.7% expressed fear of death while working.<br>❖ 26.6% felt unwelcomed by their community (stigmatisation).<br>❖ Female respondents had higher prevalence of personal (41.3% vs. 48.6%) and work-related burnout (25.0% vs. 29.1%) (p < 0.01).<br>❖ Burnout prevalence among doctors, nurses, paramedics, and administrative staff was similar. |
| 19. | Das A et al. [71] | COVID-19Depression/stress/workload | Frontline doctors involved in clinical services in OPDs, designated COVID-19 wards, screening blocks, fever clinics, and intensive care units | 422 | Tertiary care hospitals in India | *Impact on Healthcare Providers*<br>❖ Depression prevalence: 63.5% among frontline COVID-19 doctors.<br>❖ Stress prevalence: 45% among frontline COVID-19 doctors.<br>❖ Moderately severe depression: 14.2% of doctors.<br>❖ Severe depression: 3.8% of doctors.<br>❖ Risk factor for moderate or severe perceived stress: Working $\geq$ 6 h/day (adjusted odds ratio: 3.5; 95% CI: 1.9–6.3; p < 0.0001).<br>❖ Risk factors for moderate, moderately severe, or severe depression: Single relationship status (adjusted odds ratio: 2.9; 95% CI: 1.5–5.9; p = 0.002) and working $\geq$ 6 h/day (adjusted odds ratio: 10.3; 95% CI: 4.3–24.6; p < 0.0001). |
| 20. | Venkataram T et al. [72] | Neurosurgery-Stress/ financial impact | Practising neurosurgeons | 201 | Online nationwide survey | *Availability, provision, and delivery of health services*<br>❖ A significant drop of 76.25% was observed in OPD patients, and 70.59% fewer surgeries were performed during the pandemic.<br>*Adaptations and Changes*<br>**General Precautionary and Infection Prevention Measures:**<br>❖ Some respondents did not use drills (31%), and 22% wore two gowns during craniotomy procedures.<br>❖ Around 58% of the respondents had a separate ICU for COVID-19-suspect patients.<br>**Protocols and Guidelines:**<br>❖ Urgent need for evidence-based protocols.<br>**Staff management, allocation, and training:**<br>❖ 77% of the departments worked with reduced staffing during the pandemic.<br>❖ The research work of 53% of respondents was affected by the pandemic, with teaching professionals (71%) being more affected than non-teaching professionals (25%). |

Table 1. *Cont.*

| S. No. | Author | Area of Work | Participants | No. of Participants | Setting/Context | Main Outcome Measures and Findings from the Included Studies Aligned with the Review's Objectives |
|---|---|---|---|---|---|---|
| | | | | | | **PPE:**<br>❖ Need for better-quality PPE kits in adequate numbers<br>❖ 93% preferred to operate with PPE on patients with negative COVID-19 test but having high-risk features of COVID-19.<br>❖ Almost 34% of the neurosurgeons had not operated in PPE<br>❖ Of those who had, most (42%) felt that wearing PPE adversely affected their surgical performance.<br>❖ In neurosurgery where the surgeon's finesse and concentration are crucial to the surgical outcome, comfortable PPE is much needed.<br>❖ There was considerable variation in the use of protective gear by ICU staff among the respondents.<br>❖ Only 47% of the respondents' ICU staff were provided with PPE.<br><br>**Preoperative/OT/Post-operative Measures:**<br>❖ There was no consensus among neurosurgeons regarding the number of negative COVID-19 tests required before surgery.<br>❖ Before an emergency surgery, most participants (63%) did not perform any COVID-19 testing, while 35% and 2% performed 1 and 2 tests, respectively.<br>❖ Before an elective surgery, most participants (65%) required one negative COVID-19 test, while 23% required two negative tests.<br><br>**Infrastructure and resources:**<br>❖ 58% respondents had a separate ICU for their COVID-19-suspect patients.<br>❖ 19% had switched over to telemedicine,<br><br>***Impact on Healthcare Providers***<br>❖ Financial impact: 82% experienced adverse financial effects due to the pandemic.<br>❖ Financial burden: Private practitioners and those with multiple affiliations were more affected than those in government jobs (p = 0.000).<br>❖ Work-related stress: 50% reported increased work-related stress. |
| 21. | Bhandoria G et al. [73] | Gynaecological oncology | Gynaecological oncologists | 567 | The Association of Gynaecological Oncologists of India (AGOI). | ***Availability, provision, and delivery of health services***<br>❖ 44% of respondents were seeing new cancer cases in accordance with institutional guidelines, 35% were deferring these cases, and 21% had not received any new cases at the time of the survey.<br>❖ About 80% of respondents strongly believed in the cessation of elective benign surgeries, and 70% supported the cessation of cancer surgeries.<br>❖ Diagnostic services were not favoured for postponement by 68% of respondents, and 71% believed elective benign surgeries should be postponed as long as necessary to divert resources for COVID-19 care.<br>❖ 54% of respondents were not seeing patients with more than five years of disease-free survival currently. |

| S. No. | Author | Area of Work | Participants | No. of Participants | Setting/Context | Main Outcome Measures and Findings from the Included Studies Aligned with the Review's Objectives |
|---|---|---|---|---|---|---|
| | | | | | | *Adaptations and Changes*<br>**Protocol and Guidelines:**<br>❖ 75% of respondents followed institutional or national guidelines, while the rest (25%) followed international guidelines.<br>❖ Most respondents (83–92%) continued to treat advanced-stage gynaecological cancers, modifying standard management as per institutional protocols.<br>❖ 60% of respondents expressed a lack of scientific evidence among the guidelines.<br><br>**PPE:**<br>❖ 75% of respondents suggested using PPE for suspected/confirmed COVID-19 patients.<br>❖ 22% advocated for universal PPE use by all healthcare workers during healthcare delivery<br>❖ 92% believed that a combination of social distancing, face masks, and hand hygiene was an effective means of protection.<br><br>**Preoperative/OT/post-operative measures:**<br>❖ Cervical cancer: 2/3 with standard therapy, 1/3 with neoadjuvant chemo.<br>❖ Early endometrial cancer: 50% had surgery, others delayed with hormonal/NACT therapy.<br>❖ Early vulvar cancer: Most observed, few had surgery or neoadjuvant therapy.<br>❖ Mixed response to HCQ prophylaxis due to limited evidence and availability.<br><br>**Infrastructure and resources:**<br>❖ Tele-consultation services were reported to have been started by 58% HCPs |
| 22. | Babu N et al. [74] | Ophthalmology | Patients presenting to the hospital during COVID-19 lockdown | 3434 | Tertiary care dedicated ophthalmic hospital in Tamil Nadu | *Availability, provision, and delivery of health services*<br>❖ The total number of OPD visits during the lockdown period was 3434 (average 85.8 visits per day), significantly decreased compared to the same period last year (102,262 visits, average 2556.6 visits per day), indicating a 96.6% reduction in OPD volume.<br>❖ All elective surgeries were postponed during the lockdown, and emergency intervention was advised for 194 patients at imminent risk of visual loss.<br><br>*Adaptations and Changes*<br>**General Precautionary and Infection Prevention Measures:**<br>❖ Entry point screening and triaging of patients were implemented, along with recommended social distancing norms in waiting halls, OPD, and IPD to minimise crowding and patient time spent in the hospital.<br><br>**Protocols and Guidelines:**<br>❖ All the guidelines and protocols advised by All India Ophthalmological Society (AIOS) were strictly followed by all healthcare workers (HCWs) at all times.<br><br>**Staff management, allocation, and training:**<br>❖ The workforce was reduced during the lockdown period. |

| S. No. | Author | Area of Work | Participants | No. of Participants | Setting/Context | Main Outcome Measures and Findings from the Included Studies Aligned with the Review's Objectives |
|---|---|---|---|---|---|---|
| | | | | | | **PPE:**<br>❖ Adequate personal protective equipment (PPE) was provided for on-duty staff.<br><br>**Infrastructure and resources:**<br>❖ Patients were encouraged to use video and audio-based teleconsultations instead of physically coming to the hospitals for ocular complaints.<br><br>*Impact on Healthcare Seekers*<br>❖ Logistical issues during the lockdown prevented patients from receiving necessary penetrating keratoplasty<br>❖ Donor cornea shortages. |
| 23. | Khanna RC et al. [75] | Ophthalmology-Depression/fear/-Financial impact/stigma | Ophthalmologists and ophthalmology trainees | 2355 | Online nationwide survey | *Impact on Healthcare Providers*<br>❖ Depression prevalence: 33% among participants, much higher than the 10% prevalence in the general population in India.<br>❖ 53% believed that COVID-19 would significantly or seriously impact their training or profession.<br>❖ Financial implications: 37% faced difficulty in meeting living expenses.<br>❖ Depression was significantly related to concerns such as limitations in training and job security, fear of COVID-19, limited knowledge and availability of PPE, lack of adequate care in hospitals and shortage of ventilators and ICU beds, fear of carrying infection to family members.<br>❖ There is also stigmatisation targeting HCPs. |
| 24. | Chatterjee SS et al. [76] | COVID-19 Stress/anxiety/stigma | Doctors | 152 | Online survey—West Bengal | *Adaptations and Changes*<br>**General Precautionary and Infection Prevention Measures:**<br>❖ 95.4% of the doctors are practicing hand hygiene regularly, with the majority using soap and 70% alcohol-based sanitizer.<br><br>**PPE:**<br>❖ Majority of the doctors are using surgical masks (58.6%) and only a few are using N95 masks (19.7%).<br>❖ About 47.4% of them use masks for 2–6 h.<br>❖ Only 24.3% have access to PPE in their setup, and 11.2% are actually using it.<br><br>**Staff management, allocation, and training:**<br>❖ Approximately 9.2% of the doctors are currently in quarantine. |

<p style="text-align: center;">**Table 1.** *Cont.*</p>

| S. No. | Author | Area of Work | Participants | No. of Participants | Setting/Context | Main Outcome Measures and Findings from the Included Studies Aligned with the Review's Objectives |
|---|---|---|---|---|---|---|
| | | | | | | *Impact on Healthcare Providers*<br>❖ 21% experienced social ostracization due to their work in hospitals.<br>❖ 35% experienced depression, with varying severity levels.<br>❖ Involvement in high-risk procedures and working in fever clinics associated with higher odds of depression.<br>❖ Feeling ostracised also had a significant association with depression.<br>❖ 40% of doctors had some form of anxiety, and 33% experienced stress.<br>❖ Doctors performing high-risk duties (e.g., fever clinics, isolation ward) had higher psychiatric morbidity, with 63% falling into this category. |
| 25. | Subbian A et al. [28] | Gynaecologic oncology | Healthcare professionals involved in the care of gynaecologic cancer patients | 153 | National online survey | *Availability, provision, and delivery of health services*<br>❖ There was a significant reduction in surgical volume in government institutions compared to private institutions during the pandemic.<br>❖ A comparison between high and low COVID-19 incident states showed a reduction in gynaecologic cancer patient load across the country.<br><br>*Adaptations and Changes*<br>**Protocols and Guidelines**<br>❖ Cervical and vulval cancer management remained the same, but radiotherapy protocols were modified by most<br><br>**Preoperative/OT/Post-operative Measures:**<br>❖ Endometrial cancers saw a shift from minimal access surgery to conventional surgery.<br>❖ Advanced ovarian cancer was mostly managed by neoadjuvant chemotherapy.<br>❖ 93% of surgeons used additional protective measures in the OT, while only 4% used full personal protective equipment.<br>❖ 42% of surgeons used smoke evacuators during surgery. |
| 26. | Mahajan NN et al. [77] | Maternal health and Obstetrics | Obstetric patients | 600 | Multispecialty tertiary care centre in Mumbai | *Procedures and Services*<br>❖ Initially, outpatient services were functional with IPC measures and precautions, but as lockdown rules tightened, only emergency services were continued, and elective surgical procedures were delayed.<br><br>*Adaptations and Changes*<br>**General precautionary and infection prevention measures:**<br>❖ Efficient triage and screening processes were implemented at the hospital entrance for patients with COVID-19.<br>❖ Separate maternity and neonatal units were created for confirmed and suspected COVID-19 cases.<br>❖ Oxygen capacity was increased in response to increased consumption during the pandemic. |

**Table 1.** *Cont.*

| S. No. | Author | Area of Work | Participants | No. of Participants | Setting/Context | Main Outcome Measures and Findings from the Included Studies Aligned with the Review's Objectives |
|---|---|---|---|---|---|---|
| | | | | | | **Protocols and Guidelines:**<br>❖ Inpatient management protocols for antenatal and intrapartum patients were formulated following national and international guidelines.<br>❖ OT protocols, communication pathways, and transport protocols were also developed based on these guidelines.<br><br>**Staff management, allocation, and training:**<br>❖ The workforce was reduced during the lockdown period.<br>❖ Administrative teams were divided into four parts, addressing on-site changes, personnel operations, human resources, and obstetric services.<br>❖ Collaboration with transport authorities provided transport for employees due to suspended train services during the lockdown.<br>❖ Many healthcare workers were quarantined due to exposure to COVID-19.<br><br>**PPE:**<br>❖ 34.5% of practitioners used full PPE.<br>❖ Quality issues<br>❖ Operating in PPE was inconvenient due to the constant misting of eyewear/face shields and the extreme heat/dehydration.<br>❖ Procuring maximum PPE kits from administrative hospital funds or through donor liaison<br><br>**Infrastructure and resources**<br>❖ A separate maternity ward and an independent operation theatre (OT) for COVID-19 patients and suspects<br>❖ The oxygen capacity of the hospital was increased.<br>❖ Respiratory support devices were procured.<br>❖ Keeping beds at a minimum distance of 1.5 m from each other.<br>❖ Creating areas for donning, doffing, and safe and unsafe zones<br>❖ Rapid discharge policy to limit number of admitted patients and ensure capacity to support patients<br><br>**Impact of the pandemic on deliveries**<br>❖ The hospital successfully managed over 600 obstetric patients, conducting 412 deliveries, with 100 COVID-19 deliveries within a month of starting dedicated COVID-19 maternity services. |
| 27. | Moorthy RK et al. [78] | Neurosurgery | Neurosurgeons | 244 | Online survey of members of the Neurological Society of India | *Availability, provision, and delivery of health services*<br>❖ 84.3% of respondents performed semi-emergency or emergency procedures only during the pandemic period.<br>❖ 230 respondents had performed surgeries during this time. |

**Table 1.** *Cont.*

| S. No. | Author | Area of Work | Participants | No. of Participants | Setting/Context | Main Outcome Measures and Findings from the Included Studies Aligned with the Review's Objectives |
|---|---|---|---|---|---|---|
| | | | | | | *Adaptations and Changes* <br> **PPE:** <br> ❖ Concerns regarding the quality of PPE <br> ❖ Challenging to ensure quality control of PPEs being supplied by various commercial sources <br> ❖ Only 83% used N95 masks while performing surgical procedures. <br> ❖ Nearly 40% perceived that the PPE used by them was not adequately protective. <br> ❖ Some respondents (2.1%) used only a triple-layer surgical mask, and 2.6% used only a standard surgical gown during surgeries. <br> ❖ Tolerance limit not more than 2–4 h due to excessive perspiration and difficulty in breathing. <br><br> **Preoperative/OT/Post-operative Measures:** <br> ❖ 53.7% of institutions tested asymptomatic individuals for SARS-CoV-2 infection before admission to the ward/ICU. <br> ❖ Over 85% believed that preoperative testing and screening of asymptomatic individuals could reduce the risk of in-hospital transmission of the virus among healthcare workers. <br><br> *Impact on Healthcare Providers* <br> ❖ Knowledge gap and anxiety among HCWs <br> ❖ HCWs and neurosurgeons in particular are concerned about acquiring the infection due to the poor quality or lack of appropriate PPE. |
| 28. | Verma A et al. [79] | Diabetes mellitus | Patients with T1DM | 52 | Tertiary care teaching hospital | *Impact on Healthcare Seekers* <br> ❖ Treatment/doses/compliance <br> ❖ 26.9% missed insulin doses and 38.5% did not monitor blood glucose. <br> ❖ 17.4% were non-compliant with the diet during lockdown. <br> ❖ 36.5% decreased physical activity, mainly adolescents and adults. <br> ❖ 36.5% experienced hyperglycaemic episodes, with 7.7% developing DKA and requiring hospitalisation. <br> ❖ Medical condition—risen levels of blood glucose and HbA1C <br> ❖ Pre-lockdown average blood glucose: 212.3 ± 57.9 mg/dL <br> ❖ Lockdown average blood glucose: 276.9 ± 64.7 mg/dL. <br> ❖ Pre-lockdown HbA1c: 8.8 ± 1.3% (73 mmol/mol), <br> ❖ Lockdown HbA1c: 10 ± 1.5% (86 mmol/mol). <br> ❖ Medicine unavailability—challenges accessing insulin injections due to non-availability. <br> ❖ Financial burden—Some could not obtain insulin due to financial constraints |
| 29. | Joshi R et al. [80] | Diabetes mellitus | Individuals with diabetes who needed the follow up consultation | 103 | Telemedicine facility of All India Institute of Medical Sciences Bhopal | *Availability, provision, and delivery of health services* <br> ❖ Patients received dietary and lifestyle advice, adherence reinforcement, and therapeutic adjustments if needed. |

**Table 1.** *Cont.*

| S. No. | Author | Area of Work | Participants | No. of Participants | Setting/Context | Main Outcome Measures and Findings from the Included Studies Aligned with the Review's Objectives |
|---|---|---|---|---|---|---|
| | | | | | | *Adaptations and Changes*<br>**Protocols and Guidelines:**<br>❖ The treatment protocol for therapy modifications (escalations, de-escalations) was formalised at the centre and distributed to all team members following discussions.<br>**Precautionary measures:**<br>❖ Emphasised COVID-19 precautions such as hand washing, social distancing, and quarantine.<br>❖ Provided practical advice for diabetes self-management during emergencies.<br>❖ Staff management, allocation, and training:<br>❖ Utilised telemedicine with a team of trained para-clinical doctors to efficiently deliver patient care during COVID-19 lockdown restrictions.<br>❖ Positive feedback for addressing concerns and answering COVID-19 queries.<br>**Infrastructure and resources:**<br>❖ Telephonic follow up consults were given. |
| 30. | Jain A et al. [81] | Trauma/injuries | Trauma victims presented to trauma centre. | 299 | A tertiary care hospital with level 1 trauma centre and a multidisciplinary 600-bed public hospital in Delhi NCR | *Availability, provision, and delivery of health services*<br>❖ Significant decrease in the number of injured patients during the national lockdown compared to the same period in the previous year.<br>❖ An increase in the number of injured patients was observed during the 3rd and 4th phases of lockdown when liquor shops were allowed to open in Delhi.<br>*Adaptations and Changes*<br>**Protocols and Guidelines:**<br>❖ Comprehensive history and examination of all patients were conducted, and treatment followed the ATLS (Advanced Trauma Life Support) protocol. |
| 31. | Mitra M et al. [82] | Cancer care | Cancer patients in different stages of treatment and follow-up | 100 | A 600-bed tertiary care multispecialty hospital | *Availability, provision, and delivery of health services*<br>❖ Chemotherapy slot availability was problematic for 56% of respondents.<br>❖ Radiotherapy waiting hours and appointment delays were issues for 22% of respondents.<br>❖ Surgery deferral was a concern for 14% of respondents.<br>❖ Nutritionist advice delays affected 89% of respondents. |

**Table 1.** *Cont.*

| S. No. | Author | Area of Work | Participants | No. of Participants | Setting/Context | Main Outcome Measures and Findings from the Included Studies Aligned with the Review's Objectives |
|--------|--------|--------------|--------------|---------------------|-----------------|---------------------------------------------------------------------------------------------------|
| | | | | | | *Adaptations and Changes* <br> **Staff management, allocation, and training:** <br> ❖ Core COVID-19 action group and staff rotations were implemented to avoid mass quarantines. <br> ❖ Hospital bus services with social distancing were provided for staff during lockdown. <br><br> **Preoperative/OT/Post-operative Measures:** <br> ❖ Surgeries allowed only with negative COVID-19 test results. <br> ❖ Average delay in surgery: 3.22 days ($\pm$0.26) due to COVID-19 test result availability. <br> ❖ Several hospitals without dedicated COVID-19 operation theatres were not allowing surgeries for patients from containment zones. <br><br> **Infrastructure and resources:** <br> ❖ Conducting consultations through telemedicine facilities <br><br> *Impact on Healthcare Seekers* <br> ❖ Transportation/Logistics: Transportation problems from residence to the hospital were faced by 77.8% of respondents. <br> ❖ Lockdown challenges: All respondents (100%) faced more problems during the early phases of the lockdown in March and April until the healthcare system became organised. <br> ❖ Lack of guidance: Respondents mentioned a dearth of guidance from healthcare personnel on accessing care. <br> ❖ Unavailability of peer support and counselling was problematic for 94% of respondents. <br> ❖ Availability of chemotherapy medications was an issue for 22% of respondents. <br> ❖ Slot availability for teleconsultation was an issue for 42% of respondents. <br> ❖ Increased anxiety: 92% experienced heightened anxiety levels. <br> ❖ Visitor restrictions affected 72% of respondents. <br> ❖ Difficulty in maintaining precautions was an issue for 33.3% of respondents. <br> ❖ Reasons for increased anxiety in cancer patients: Fear of COVID-19 infection, concern over delayed treatment and disease progression, suboptimal treatment, fear of death, and financial crisis for family members. |
| 32. | Ghosh J et al. [83] | Oncology | All patients age $\geq$ 18 years who are actively undergoing systemic therapy for solid malignancies | 302 | Department of Medical Oncology | *Impact on Healthcare Seekers* <br> ❖ Patient preference: 68% of patients wanted to continue chemotherapy despite the pandemic, prioritising cancer treatment over concerns about SARS-CoV-2. <br> ❖ Concerns about treatment: About two-thirds of patients were bothered by the potential deferral of chemotherapy or visiting hospitals during the pandemic. <br> ❖ Concerns about cancer progression: Approximately one-third of patients were worried about the possibility of cancer progression if their therapy was hindered, while around 70% expressed concerns about cancer progression if chemotherapy was stopped. |

Table 1. *Cont.*

| S. No. | Author | Area of Work | Participants | No. of Participants | Setting/Context | Main Outcome Measures and Findings from the Included Studies Aligned with the Review's Objectives |
|---|---|---|---|---|---|---|
| 33. | Panda PK et al. [84] | Neurology | Caregivers of children suffering from neurological disorders | 153 | Paediatric Neurology Division, All India Institute of Medical Sciences, Rishikesh | *Adaptations and Changes* <br>**Infrastructure and resources:** <br>❖ Telecommunication successfully used for prescribing and modifying AEDs in children with epilepsy. <br>❖ 96% of caregivers were satisfied with the medical advice. <br>❖ 26% of caregivers inquired about COVID-19 symptoms and risks. <br>❖ 7% of caregivers contacted for missed scheduled visits without active health issues. |
| 34. | Nachimuthu S et al. [85] | Diabetes mellitus | Diabetic patients | 100 | A diabetes speciality hospital in Chennai | *Impact on Healthcare Seekers* <br>❖ Only 28% of participants tested their blood sugar levels regularly. <br>❖ SMBG machines/strips unavailability or limited practice of SMBG in India could be reasons. <br>❖ Exercise and Diet Control: <br>❖ 80% of the study population maintained regular exercise and diet control. <br>❖ Physical activity was adapted within their homes and apartments due to limitations on going for walks. <br>❖ Anxiety: 40% of the population expressed anxiety about the COVID-19 situation. |
| 35. | Prasad N et al. [86] | Kidney diseases | Director or the heads of the departments | 2517 | Public sector tertiary care teaching institutes and private sector corporate hospitals | *Availability, provision, and delivery of health services* <br>❖ Serving dialysis stations declined from 523 to 496 after lockdown (9 in public hospitals, 18 in private). <br>❖ Total dialysis patients reduced in these centres. <br>❖ Renal transplants decreased. <br>❖ 28.2% of patients missed one or more dialysis sessions. <br>❖ 2.74% required emergency dialysis sessions. <br>❖ 4.13% stopped reporting for dialysis, and 0.36% confirmed deaths. <br>❖ Outpatient attendance reduced by 92.3%, and inpatient service by 61% in surveyed hospitals. <br><br>*Adaptations and Changes* <br>❖ Heterogeneity in testing for COVID-19, with some hospitals adhering to the advisory, whereas others used their own protocols. <br><br>**Staff management, allocation, and training:** <br>❖ Many kidney doctors were quarantined due to contact exposure to COVID-19 patients. <br>❖ Different policies for testing and self-isolation in public and private hospitals. <br><br>**Infrastructure and resources:** <br>❖ 12 centres (63%) had created cohorting solutions for dialysing COVID-19-positive or COVID-19-suspect patients <br>❖ Majority (67%). adopted isolation rooms with dedicated machines for dialysis <br>❖ Tele-consultation started but accessed by only a small number of patients. |

| S. No. | Author | Area of Work | Participants | No. of Participants | Setting/Context | Main Outcome Measures and Findings from the Included Studies Aligned with the Review's Objectives |
|---|---|---|---|---|---|---|
| | | | | | | ***Impact on Healthcare Providers***<br>❖ Doctor quarantines: Many doctors were quarantined due to incidental exposure.<br>❖ Non-physician staff challenges: Technicians and nurses were more likely to abstain from work due to a combination of ignorance, fear of infection, and transportation difficulties during the lockdown.<br><br>***Impact on Healthcare Seekers***<br>❖ Impact on dialysis patients: Patients in public sector hospitals were more likely to miss dialysis or drop out entirely.<br>❖ Access to renal care: The conversion of public hospitals into COVID-19 care centres significantly affected access to renal care, particularly for those from low-middle socioeconomic groups who rely on the public sector for healthcare services. |
| 36. | Ghosal S et al. [87] | Diabetes mellitus | Non-diabetic household members of T2D patients | 100 | Tertiary care diabetes centre | ***Impact on Healthcare Seekers***<br>❖ 40% of individuals gained weight, with 16% gaining between 2.0 and 5.0 kg. The diabetes risk score increased in 7% of the population. The percentage of individuals with a BMI greater than 30 kg/m$^2$ also increased from 18% to 21%. |
| 37. | Chopra S et al. [88] | Lifestyle related behaviours | Adults age $\geq$ 18 years | 995 | Nationwide online survey | ***Impact on Healthcare Seekers***<br>❖ Physical activity changes: Decline in moderate-intensity aerobic exercises; some engagement in walking, at-home workouts, and yoga, but overall reduced physical activity.<br>❖ Increased stress levels during COVID-19.<br>❖ Dietary changes during COVID-19: Increased regular meal consumption and balanced diet; decreased intake of unhealthy food and sugary beverages.<br>❖ Factors influencing dietary improvements: Fear of COVID-19, preference for home-cooked food, and reduced eating out and socialising.<br>❖ Factors contributing to reduced physical activity: Lack of motivation, limited time availability, and restricted access to fitness facilities.<br>❖ Reasons for increased stress and anxiety: Fear of infection, concerns for family, boredom/loneliness, and financial loss. |
| 38. | Nilima N et al. [26] | Community/individual health | People from all the states | 1316 | Nationwide online survey | ***Impact on Healthcare Seekers***<br>❖ Family health and lockdown compliance: Worrying about family health was associated with higher adherence to lockdown measures.<br>❖ Relationship between non-compliance and dissatisfaction: Among those who did not follow the measures, a significant portion (37.5%) expressed dissatisfaction with the government's strategy. |

The participants involved in these studies encompassed healthcare providers, including surgeons, doctors, nurses, and frontline workers as well as healthcare seekers, specifically patients. The findings derived from these studies have been meticulously analysed, interpreted, and categorised based on specific thematic areas.

3.2.1. Impact of COVID-19 and Turn of Events on Provision, Availability, and Utilisation of Health Services

(a)    Outpatient department (OPD) services

OPD services were described in 19 studies [54–59,61,62,64,65,68,69,72,74,77,79,80,82,86] involving 11,890 healthcare stakeholders. These studies covered various fields, such as oncology, neurosurgery, ophthalmology, maternal health, primary health care, general surgery, orthopaedics, and palliative care. All 19 studies reported a significant reduction in OPD services during the lockdown phase (March–May 2020), which persisted for several months thereafter. On further analysis, the reduction in these OPD services was statistically significant ($p < 0.05$ or $<0.01$), as reported in two studies [59,72]. Another study reported a staggering 97% decrease in OPD visits compared to the corresponding period in 2019 [74].

Among the OPD clinics at Primary Health Care sites, the greatest reduction was observed in noncommunicable diseases (NCDs) and immunisation clinics, while antenatal care services (ANCs) experienced lesser disruptions. General OPDs were the least affected [58].

Several included studies also reported the establishment and existence of Screening OPDs [54,55,57,74,77], where incoming patients were assessed for COVID-19-related symptoms and their travel history and underwent thermal screening and subsequent COVID-19 testing.

(b)    Elective services

Elective procedures are described in 16 studies [28,54–56,59–65,67,72–74,77] involving a total of 9268 subjects. These studies consistently report a significant impact on elective procedures across various areas of healthcare services, with procedures either being completely halted, significantly reduced, or deferred.

For instance, several studies focus on cancer care, all indicating a noteworthy decline, deferral, or even cessation of elective oncological procedures [28,54,62,64,73]. These procedures included radiotherapy [28,62], surgery, chemotherapy [62], tumour boards [28], and nutritionist consultations [28]. In cases where appropriate, such as advanced cases, deferrals were made to manage cancer through neoadjuvant chemotherapy [28,54].

Orthopaedics also experienced a similar trend, with the majority of elective surgeries being halted [63]. Many orthopaedic surgeons limited their practice to performing only essential trauma surgeries (62%) or ceased surgeries altogether (35%) during this period. Likewise, 93% of general surgical practices ceased all elective surgical procedures [59].

There was a significant decrease in vitreoretinal ophthalmological surgeries [67], with many of them being postponed [74]. Neurosurgery also witnessed a sharp decline, with approximately 70% fewer surgeries performed ($p = 0.000$) [72].

In the realm of maternal health services, a substantial 45% decrease in the number of deliveries was reported compared to pre-COVID-19 times, and this decline was statistically significant ($p < 0.001$) [61]. Furthermore, there were notable delays in service provision [77].

When it comes to cardiovascular diseases, there was a significant reduction in STEMI (ST-Segment Elevation Myocardial Infarction) admissions by 67% during the lockdown period compared to the pre-COVID-19 period. The reduction in NSTEMI (Non-ST-segment elevated myocardial infarction) cases was even more significant, reaching 93% within the same timeframe [60].

(c)    Emergency services

Emergency services are the subject of 20 studies [28,54–57,59,61,62,64,67,68,72–74,77,78, 80,81,84,86] involving 12,850 healthcare stakeholders. These services encompass various

specialities such as ophthalmology, maternal health, non-communicable diseases (including cardiovascular diseases, diabetes, and kidney diseases), neurosurgery, orthopaedics, injuries/trauma, and general surgical practice.

Overall, the majority of healthcare providers and institutions continued to deliver emergency and urgent services, taking necessary precautions and adapting protocols and techniques accordingly. However, the provision, utilisation, and availability of these services were impacted and altered due to challenges related to accessibility, transportation difficulties, infrastructural changes, resource constraints, and concerns arising from the lockdown. Consequently, these factors resulted in suboptimal medical care in life-threatening emergencies.

The changes and altered patterns of these services are described in several studies in different clinical areas.

In ophthalmological services [56], trauma cases accounted for the majority (51.9%), and a significant portion of doctors (83%) focused solely on emergency cases. Notably, there was a sharp 60% increase in home-related trauma cases during the lockdown, particularly among children. Despite this, the number of immediate surgical interventions among children has reduced (>80%) compared to pre-COVID-19 times, possibly due to difficulty in availing transport and poor access to healthcare facilities. Another study [74] also reported that the overall volume of surgical interventions was reduced, with only a small fraction of emergency interventions continued in cases of imminent risk of visual loss.

In cardiovascular diseases [60], there was an upsurge in acute coronary syndrome (ACS) patients presenting with delayed symptoms and mechanical complications. Similarly, in the context of diabetes [80], overall emergency services were reduced, with only individuals experiencing severe hyperglycaemia and recurrent hypoglycaemia receiving emergency consultations or services. Additionally, renal diseases witnessed a decline in the number of available dialysis stations and patients, and renal transplants both in the public and private sectors [86].

In cancer care also, despite an overall reduction in the patient load for gynaecologic cancers, a significant proportion of healthcare providers (69–92%) continued to provide treatment for emergency cases and advanced-stage emergency gynaecological cancers [64,73]. They adapted their approaches according to institutional protocols [73], shifting from minimal access surgery to conventional surgery, implementing neoadjuvant chemotherapy, and modifying radiotherapy protocols.

In obstetrics, there has been an increase in the volume of maternal and obstetric emergencies due to factors such as the reduced number of antenatal visits, delays in accessing services (due to travel restrictions), and waiting until the last moment due to infection fears [61].

In orthopaedics, approximately 21% had even ceased emergency surgeries entirely, and many surgeons were exclusively performing unavoidable emergency trauma surgeries (62%) [63].

3.2.2. Health System's Response—Adaptations and Efforts for Resumption of Health Care Services

(a)    General precautionary and infection prevention measures

Ten studies [54,57,58,67,69,72,74,76,77,80] involving 5911 healthcare providers/seekers examined general precautionary and infection prevention measures.

There had been widespread implementation of some of the essential general precautionary and infection prevention measures, such as initial screening at entry, mandatory mask-wearing, and limitations on the number of attendants visiting or accompanying the patients [54,57,67,69,74,77].

Furthermore, additional infection prevention measures were implemented at various study sites [54,57,67,69,74,80]. These included providing hand sanitizers at accessible locations and displaying audio-visual notices at regular intervals to promote hand washing, mask usage, and adherence to social distancing norms. Furthermore, standing spaces were

marked to maintain physical distance, adequately spaced sitting areas were designated, and surfaces were regularly sanitized.

Healthcare providers were reported to diligently implement infection prevention measures and take necessary precautions while delivering services. For instance, they frequently practiced hand sanitization using alcohol-based sanitizers, wore N95 masks, face shields, or safety goggles, utilised disposable gowns and gloves, employed double gowns when necessary, and set up dedicated ICUs for suspected COVID-19 cases [54,57,69,72,76,80].

While these precautionary measures were generally adhered to in most hospitals, they were significantly compromised at numerous primary healthcare centres [58]. For example, 76% of the sites lacked airborne infection control measures, 24% lacked adequate handwashing facilities for patients, and patient queuing averaged 14.1 individuals at many centres. Nevertheless, chemical disinfection of the PHCs was being undertaken at most (82%) sites with daily, alternate day, and less frequent disinfection reportedly conducted in 53%, 14%, and 20% of the sites, respectively.

(b)    Protocol and guidelines

To guide the operations, healthcare centres, hospitals, and clinics implemented various guidelines and protocols. The adoption and adherence to these diverse protocols and guidelines are discussed in thirteen studies [28,54,59,62,64,67,72–74,77,80,81,86], encompassing 8697 participants. Some institutions formulated or modified their guidelines based on existing literature and guidelines, while others followed state, national, or international guidelines.

For example, in two oncological studies [64,73], it was found that a significant majority of institutes and surgeons (69–75%) followed institutional or national guidelines, while the remaining (25% to 31%) adhered to international or alternative guidelines. This heterogeneity in the adoption and adherence to guidelines and protocols was further confirmed by a survey [86] conducted in 19 major hospitals, which reported variations among facilities in terms of adhering to the guidelines for testing SARS-CoV-2 issued by the Ministry of Health and Family Welfare (MoHFW) and the Indian Council of Medical Research (ICMR), with some institutions using their own protocols.

In another two oncological studies, cancer care hospitals [54,62] developed and implemented their own comprehensive protocols for all areas of work—general hospital, staff, preoperative, OT, postoperative, and emergency procedures—based on the limited available literature at the beginning of the pandemic. Similarly, a tertiary care centre formalised a diabetes treatment protocol for therapy modifications (escalations, de-escalations) and subsequently distributed it to all team members [80].

In obstetrics, a hospital formulated inpatient management protocols for both antenatal and intrapartum patients based on the national and international guidelines from organisations such as the International Federation of Gynaecology and Obstetrics (FIGO), the Centres for Disease Control and Prevention (CDC), and the World Health Organisation (WHO) [77]. These protocols encompassed various aspects, including communication pathways and transport protocols in the operating theatre. Many others modified or adapted the guidelines and protocols in different areas in an effort to continue operations [28,67,81]. On the other hand, in an ophthalmological study, surgeons kept adhering to the guidelines and precautions advised by The All India Ophthalmological Society (AIOS) [74].

After the initial phase, within 3–4 months of the pandemic (May–June 2020), the majority of healthcare facilities had implemented protocols for reopening services in the "new normal" situation. However, despite the availability of these protocols and guidelines, a significant proportion (60%) of healthcare providers expressed a lack of scientific evidence supporting the guidelines [54,62,72,73,86]. This absence of specific protocols also contributed to the stress experienced by healthcare workers, highlighting the urgent need for evidence-based protocols [54,72]. For example, a significant majority (71.5%) of surgeons expressed the explicit need for guidelines addressing safety measures for future surgical practices and providing solutions to mitigate the aftereffects of the pandemic [59].

(c)    Staff allocation, management, and training

Sixteen (*n* = 16) studies [54,55,57,58,62,66–68,70,72,74,76,77,80,82,86] involving 11,408 healthcare stakeholders provide valuable insights into staff allocation, training, and management during the peak of the pandemic. These studies shed light on the various measures implemented to enhance the efficiency of human resources and prevent the transmission of infection.

To effectively respond to the crisis, healthcare facilities established initiatives such as creating a dedicated COVID-19 action group, implementing staff rotation with different teams working in shifts, and dividing the workforce into separate groups for COVID-19 and non-COVID-19 patients [54,62,67,82]. Some studies also highlighted the adoption of reduced staffing strategies to preserve the workforce and prevent fatigue in hospitals and clinics [55,67,72,74]. A study in neurosurgical department revealed that all out-station leaves of healthcare workers were suspended to prevent community spread and ensure the maximum availability of the workforce [55]. Physical meetings in the hospitals were either cancelled or replaced by virtual meetings [54,55] and remote work was authorised for clinical, resident, and support staff [55]. Staff members arriving from areas declared as containment zones were granted paid leaves to mitigate the risk of cross-infection [54]. Furthermore, operation theatres were observed to operate with minimal staff, organised into teams or shifts [67].

Despite these measures, there have been instances of human resource constraints reported to have weighed heavily on the teams when vulnerable (older, pregnant) staff were relieved from high-risk duties [66]. The existing staff faced the challenge of balancing their time between designing and implementing new interventions and providing essential care.

Furthermore, quarantining staff as per infection prevention guidelines also contributed to the availability of lesser staff at any given point. Both private and public healthcare providers underwent adequate quarantine measures in cases of incidental exposure or if they were working in isolation wards or had any form of exposure to COVID-19 [54,62,67,68,76,77,80,82,86]. One study [62] explains that all the staff was divided into two groups, working at different times, and strict precautions were taken to ensure no overlap between these two groups. If any staff or patient turned positive for COVID-19, the entire group was quarantined and tested.

Training and workshops played an integral role in enhancing staff efficiency, ensuring adherence to protocols, preventing COVID-19 infections among personnel, and delivering comprehensive care to both COVID-19 and non-COVID-19 patients.

As underscored in several studies [55,57,58,62,66,77,80], the training related to infection control, sanitization, PPE Kit, donning and doffing of PPE, and the management of COVID-19 patients as per protocols were made mandatory for all staff and doctors at institutions. Even in primary health centres, the training related to the safe and effective management of patients with presumptive COVID-19 had been provided at approximately 78% of the sites to the health staff [58].

In several instances, the training and meetings were conducted online, including webinars [55,70,80], and information about guidelines and management protocols for COVID-19 widely disseminated in closed groups formed, aided by social media [77].

Furthermore, a study [80] demonstrated the practical and effective approach of training and engaging para-clinical doctors in providing diabetes care and COVID-19 information through underused telemedicine. The study further recommends extending similar efforts to address other chronic conditions like hypertension and asthma by training and building the capacity of more such teams of doctors. Also, this strategy can be adapted at other resource-limited centres facing challenges in delivering patient care services for non-COVID-19 illnesses such as diabetes, cardiovascular diseases, and more during the pandemic [80].

Another study [70] reporting burnout among HCPs stresses that HCPs should be provided information, instructions, training, and technical updates on COVID-19 through frequent communication to curb fear and burnout.

(d)  Personal Protective Equipment (PPE)

This analysis encompasses 22 studies involving 15,144 participants [27,54–59,61,62,64–66,68–70,72–74,76–78,86], which delve into the use and challenges of Personal Protective Equipment (PPE) among healthcare providers (HCPs) during consultations, surgeries, and other healthcare services. While the adoption of PPE became widespread, specific instances, predicaments, and challenges have been identified.

A common-sense, rationed approach to using resources like PPE during a pandemic of this magnitude is being followed cautiously in India and worldwide. Most practitioners felt that complete PPE needed to be reserved for workers dealing with high-risk patients [58]. While this seems to be a reasonable approach, many argue that primary care providers are also known to be at an increased risk, especially during epidemics. If the allocation of PPE limits the provision of N95 masks to only those HCPs directly involved in the management of confirmed COVID-19 cases, HCPs in resource-constrained settings working in enclosed small spaces without adequate ventilation and likely overcrowded are rendered highly vulnerable to COVID-19 in the absence of adequate PPE provision. Unfortunately, only half of the medical colleges and institutions provided N95 masks to healthcare providers at their primary health care facilities. PPE suits, N-95 masks, and surgical masks were available at only 27%, 50% and 39% of primary health care sites, respectively [58].

Furthermore, the unavailability and short supply of adequate quality PPE were widespread concerns reported in a plethora of included studies [27,54,58,64,68,72,76–78,86]. Supply chain disruptions and increased demand contributed to these shortages of PPE [78,86]. In response, a major tertiary care COVID-19 hospital resorted to procuring maximum PPE kits using administrative funds or donor liaison to address the heightened demand [77].

While the health system grappled with supply and demand issues, several studies highlighted the need for better quality and additional quantities of PPE. For instance, studies involving ophthalmologists, oncologists, and neurologists indicated a preference for wearing additional PPE during surgeries [28,56,62,72] to ensure adequate protection. However, even with full PPE, medical personnel were not always spared from infections [65], raising concerns about the quality of PPE. A survey of neurosurgeons revealed that nearly 40% were dissatisfied with the available PPE's quality [78]. Another study [54] emphasised the increased need for better protective equipment, particularly amongst surgeons during operations. Additionally, an alarming 34% of surveyed neurosurgeons [72] had not operated in PPE, and among those who did, most (42%) felt it negatively impacted their surgical performance, underscoring the vital need for comfortable and high-quality PPE during complex neurosurgeries that demand significant finesse, concentration, and time. This predicament left surgeons grappling with the dilemma of either risking infection or compromising their surgical performance due to inadequate PPE [72]. Despite these quality concerns, ensuring quality control was challenging due to variations in suppliers and excessive demand [78].

Several other studies shed light on the challenges associated with donning PPE, especially during surgical procedures. Operating in PPE resulted in constant misting of eyewear/face shields [77], extreme heat/dehydration [66,77] and difficulty in breathing [78]. Respondents in different studies indicated limited tolerance periods for wearing PPE, ranging from 2 to 4 h at most [66,76,78]. In some slum clinics, health workers were provided with reusable water-resistant PPE. However, disinfecting these PPEs in the confined space of the slum clinics posed additional challenges [66].

(e)  Preoperative/OT/post-operative measures

Twelve studies [28,54–56,59,62,64,67,72,73,78,82,83] provide insights into preoperative, intraoperative, and postoperative measures, involving a total of 3814 participants. Several studies consistently emphasised the necessity of COVID-19 testing for all preoperative patients, ensuring that only those testing negative for the virus proceeded with surgeries [55,78,82]. However, at the onset of the pandemic, there was significant uncertainty and scepticism surrounding preoperative procedures. For instance, an early study [56] revealed that the

majority of clinicians (63%) were uncertain about the appropriate course of action and were eagerly awaiting guidelines regarding preoperative screening. Another study [64] highlighted a similar predicament during the initial phase, as the recommendations by the Government of India did not initially allow for preoperative COVID-19 testing, making it challenging to make decisions regarding surgery based on COVID-19 status.

Furthermore, there was no consensus among practitioners regarding the number of negative COVID-19 tests required preoperatively [72]. Before emergency surgery, 63% did not perform any testing, 35% performed 1 test, and 2% performed 2 tests. For elective surgery, 65% required one negative test, while 23% needed two negative tests [72]. This ambiguity surrounding preoperative testing and procedures was widely observed during the early stages of the pandemic.

At some institutions/clinics however, testing became an essential prerequisite criterion [55,78,82]. As a result, delays in surgeries were common, with an average delay of 3.22 days ($\pm$0.26) reported in one study [82]. Hospitals without dedicated COVID-19 operation theatres allowed surgeries only for patients with negative test results, especially those from containment zones. In another study [78], 54% of institutions performed pre-admission testing, and over 85% of respondents believed preoperative testing and screening of asymptomatic individuals were crucial to reduce in-hospital transmission among healthcare workers.

In addition to preoperative testing, certain institutions implemented changes in their surgical strategies and techniques. For instance, an ophthalmological study reported modifications in their operation theatre (OT) sterilisation protocol [67]. Furthermore, some surgeons adopted the use of a "heads-up" 3D visualisation system during vitreoretinal surgery to increase the distance between the surgeon and the patient, thereby minimising the risk of exposure. Additionally, in several settings, strict precautions were taken upon admitting patients, allowing only one attendant [54,72,82].

In the field of oncology, studies highlighted the adoption of neoadjuvant chemotherapy as an alternative to upfront surgery, deviating from the standard of care but serving as a viable option during resource-constrained times of the pandemic [28,54,62,73,83]. For instance, in one of these studies, elderly and frail patients with operable lesions, who were at a higher risk at COVID-19 infection were advised to either wait or undergo neoadjuvant therapy [62]. Furthermore, surgeries for pre-invasive lesions and slow-growing cancers were postponed, and only life-saving procedures were performed at most healthcare facilities. Other cases were either kept under observation or considered for neoadjuvant therapy, specifically for carefully selected eligible cases [28,54,62,73,83].

In the realm of neurosurgical procedures, certain adaptations were made to minimise exposure risk. Some surgeons avoided craniotomy whenever possible, while others opted not to use drills and adopted the practice of wearing two gowns, removing the outer gown after 20 min following craniotomy to minimise contamination [72]. In general surgical practice also [59], various adaptations were implemented to reduce the risk of infection, including the use of Hydroxychloroquine (HCQ) for chemoprophylaxis, considering open surgery, and using filters for desufflation.

(f)  Physical infrastructure and resources

Twenty studies [54–59,61–64,66,69,72–74,77,80,82,84,86] involving 12,409 subjects have documented various modifications made to infrastructure and resources. The primary objective of these changes was to create COVID-safe environments in buildings, wards, operating rooms (ORs), and other facilities. For instance, several studies emphasise the allocation of separate buildings, units, or wards exclusively dedicated to managing or treating COVID-19 patients [54,72,77,86]. Additionally, engineering controls, such as physical barriers, curtains, or partitions [72,77], were put in place to reduce exposure risk and contamination. Healthcare facilities also established distinct and regulated entry and exit points for both patients and healthcare providers [54,58,77], set up special screening areas near the entrance of the hospital [54,58,62,77], and designated separate areas for donning and doffing personal protective equipment (PPE) [62,77].

Efforts were also made to ensure physical distancing, with sitting and standing areas properly marked and spaced [54,58]. In addition, a minimum distance of 1.5–1.8 m was maintained between beds [62,77], and the number of patients in waiting areas, common spaces, elevators, wards, and operating rooms was limited and regulated to prevent overcrowding [54,62]. Healthcare facilities prioritised environmental cleaning, proper ventilation, air conditioning, and air filtration in areas such as wards, ICUs, and operating rooms [54,77]. For example, one study reported the maintenance of adequate air exchange rates in ORs, with the use of high-efficiency particulate air filters for ultra-filtration, strict fumigation after each case, and suction to evacuate smoke and ensure clean air and ventilation.

A health care clinic in slums [66] also improved ventilation by relocating consultation rooms to open spaces to reduce the risk of viral transmission. However, challenges persisted as patient registration, laboratory work, and dental procedures were still conducted in rooms with compromised ventilation, leading to long queues and waiting times. Another study [58] assessing primary health facilities also highlighted inadequate ventilation, suboptimal infrastructural capacity and a lack of separate entries, hindering physical distancing efforts at most PHCs.

Besides these adaptations, several changes were made in the processes. For instance, surgical procedures in operating rooms were spaced out with sufficient time intervals to ensure proper sterilisation [54,62]. Solutions were implemented to cohort COVID-positive patients or suspected cases, including the creation of isolation rooms, dedicated machines and dedicated shifts [72]. Furthermore, adjustments were made to resources, such as increasing the hospital's oxygen capacity and arranging more beds with oxygen ports and supply in response to the increased demand for oxygen during the COVID-19 pandemic [77]. Further efforts were increased for procuring ventilators, non-invasive options (HFNO and BiPAP) and other respiratory support devices [77].

Furthermore, telemedicine experienced widespread and varied adoption, enabling healthcare providers to extend their reach to patients in diverse medical specialities [55–57,59,61,63,64, 66,69,72–74,80,82,84,86]. It facilitated remote consultations, follow-ups, and monitoring, minimising the need for in-person visits and reducing the risk of exposure in varied healthcare settings. It proved to be a valuable tool in ensuring continuity of care for patients while adhering to safety measures during the pandemic.

### 3.2.3. Impact of Pandemic on Healthcare Providers and Individuals/Communities

(a) Impact on health care providers (HCPs)

 • Depression, stress, anxiety, and burnout in health care providers (HCPs)

Fourteen studies [27,54,56,58,62,65,66,68,70–72,75,76,78] involving 8568 participants examined the mental health status of HCPs during the pandemic, revealing a significant prevalence of pandemic-related stress and burnout among them.

Approximately 33% to 35% of HCPs [75,76] experienced depression, surpassing the prevalence of 10% for common mental disorders in the general population of India [89]. The mean score of pandemic-related burnout in HCPs [68] was significantly higher than both personal and work-related scores (51.37 vs. 49.7 and 39.7, respectively, $p < 0.05$). Additionally, HCPs working in high-risk hospital environments demonstrated a significantly greater prevalence of work-related (27.8% vs. 21.2%, $p < 0.01$) and pandemic-related burnout (53.9% vs. 45.6%, $p < 0.01$). Other studies have also documented varying degrees of depression, stress, anxiety, and burnout among HCPs working in different specialities [27,58,66,78].

Among front-line COVID-19 doctors [71], the prevalence of depression and stress symptoms was much higher, with rates reaching 64% and 45%, respectively. Another study [66], which focused on front-line health workers in slums, highlighted the enormous emotional and physical toll they experienced. These workers not only put in long hours fully clad in PPE but also faced additional challenges, such as clashes and violence due to limited economic activity among slum dwellers during the pandemic. Furthermore, the

population's resistance to adhering to safety measures like wearing masks and following clinic protocols added to the difficulties faced by them.

Numerous factors contributed to stress and anxiety among HCPs [27,54,56,58,62,66, 68,70–72,75,76,78,86] to varying degrees, including the fear of contracting COVID-19 due to extreme proximity during treatment, anxieties regarding the availability of sufficient PPE, and the physical strain of working long hours in protective gear. Loneliness during quarantine, difficulties in maintaining social connections, and the challenges associated with routine tasks (OR, OPD, and IPD works) also took a heavy toll on their well-being. Additionally, insufficient guidelines and knowledge gaps, limited administrative support, financial losses, and staff shortages further exacerbated the burden they experienced.

. Fear of contracting infection and carrying it at home

The fear of contracting and transmitting the infection has been addressed in nine studies [27,54,56,62,66,70,75,78,86], involving 9490 participants. Among HCPs caring for COVID-19 patients, approximately 55% expressed fear of contracting the infection themselves, while an even higher percentage (67%) feared transmitting it to their families [70].

A separate study [27] conducted across ten states and one union territory revealed that a significant majority of HCPs (78%) experienced serious apprehensions about infecting their friends or family members. Likewise, healthcare workers in the community health division of a hospital serving a large slum area [66] emphasised their primary concern of transmitting the infection to vulnerable family members, including the elderly, immune-compromised individuals, and those with chronic medical conditions.

Notably, this fear of transmitting the infection to families and close ones was even identified as a cause of depression among HCPs in some studies [27,75]. Furthermore, certain specialities [56,78], such as ophthalmology and neurology, perceived a higher risk of contracting COVID-19 while examining and treating patients, leading to the requirement for additional protective gear compared to other specialities.

. Stigmatisation

Stigmatisation targeting healthcare providers (HCPs) has been documented in five studies [62,66,70,75,76], involving 5173 participants. A significant proportion of HCPs in a comprehensive survey (26.6%) reported feeling unwelcome in their communities [70]. The studies conducted in both community and hospital settings [62,66,76] highlighted that healthcare workers were perceived as carriers of infection and often faced ostracisation from friends, neighbours, and society as a self-protective measure. This stigmatisation had negative impacts on the mental well-being of HCPs, leading to feelings of depression, stress, and anxiety [70,75,76].

. Financial impact

The financial impact of the pandemic on healthcare practitioners is documented in five studies [27,54,59,63,75], involving 3489 participants. A survey revealed that a significant majority (82%) of healthcare providers experienced financial harm due to the pandemic [72]. In another study [59] over 50% of the healthcare providers reported a decrease of more than 75% in their monthly income, while 22% faced a 50–75% reduction.

The financial burden was particularly pronounced among private practitioners and those with multiple affiliations compared to those in government jobs (*p* = 0.000) [59,63,72].

Notably, approximately 28% and 33% of respondents who owned hospitals expected their income to decline by more than 90% [63] and anticipated a monthly financial liability of $30,000 [59], respectively. Surgeons with more years of practice, specifically those with 20 to 30 years or more than 30 years of experience, were particularly affected compared to their younger counterparts [63].

Some other studies [54,63,75] revealed that the decreased income from practice poses significant implications on their financial sustainability, career, and quality of training, prompting many to consider changing their financial strategies in response to the challenges posed by the pandemic.

(b)   Effect of pandemic on healthcare seekers

Thirteen studies [26,57,61,62,69,74,79,82,83,85–88] involving 11,371 participants examined the effects of the pandemic on the physical and mental health of individuals. These studies highlighted various impacts on disease progression, disease status, lifestyle factors such as physical activity and diet, and medication adherence.

Specifically, among diabetic patients, there was an observed increase in missed insulin doses (27%) [79], irregular blood glucose monitoring (28–39%) [79,85], reduced engagement in regular physical activities (37%) [79], and decreased compliance with dietary guidelines (17%) [79]. The unavailability of insulin injections [79] and Self-Monitoring Blood Glucose (SMBG) machines/strips [85] during the lockdown was reported to have contributed to missed insulin doses and irregular blood glucose monitoring. As a result, average blood glucose and HbA1c levels were elevated, hyperglycaemic episodes increased (37%), and there was a higher incidence of Diabetic ketoacidosis (DKA) and hospitalisations (8%) [79] among these patients. Besides, an increased risk of type 2 diabetes was also observed among non-diabetic people [87] owing to weight gain and an elevated diabetes risk score.

In the case of cancer patients, they encountered numerous challenges, such as limited availability of chemotherapy slots (56%), long waiting hours despite scheduled appointments (22%), postponed surgeries (14%) and tumour board meetings (20%) [57,82,83]. Teleconsultations and nutritionist advice were also significantly delayed (42% and 89%, respectively), and there was a concerning shortage of chemotherapy medications (22%) [57,82,83]. Most of the patients (68%) wanted their chemotherapy to continue despite the pandemic, emphasising their deep concerns about cancer progression, outweighing their fear of contracting SARS-CoV-2 [83].

In obstetric complications [61], a surge in the number of high-risk pregnancies (by 7.2% points) and aggravation in one or more complications (44.7%) was observed owing to the delay in seeking health care. The reason for the delay in health-seeking was either the strict lockdown and lack of transportation or avoidance of visits due to the fear of catching infections. Anaemia was overlooked in most women, followed by pregnancy-induced hypertension. Many women opted to wait at home for labour or choose home deliveries, contributing to adverse outcomes such as eclampsia, acute renal failure, sepsis, and pneumonia.

Patients across multiple studies [61,62,79,82,83,88] expressed widespread fear due to the increased risk of infection associated with their existing conditions. They were concerned about the potential deterioration of their underlying diseases as a result of delayed or suboptimal treatment. Apart from the fear of adverse outcomes, patients also reported challenges such as limited guidance from healthcare professionals [82], shortages of essential medications and donors [69,74,79,82], escalating costs of planned medical procedures [58], higher expenses for hospital visits [57,62], and the need to travel long distances to access medical care, putting their immunocompromised state at risk [57]. These factors contributed to heightened anxiety and stress among the patients.

Furthermore, transportation and travel issues significantly impacted patients seeking treatment during the lockdown period [57,61,62,69,74]. The unavailability of public transport, limited access to fuelling stations, and the need for permissions under strict lockdown measures compounded the difficulties patients faced in reaching hospitals. Moreover, the conversion of public hospitals into COVID-19 care centres further hindered access to treatment, especially for individuals from low to middle socioeconomic backgrounds who depend on the public sector for healthcare services [86].

The fear of job loss and financial hardships [57,79,82,88], feelings of loneliness and boredom [88], the worsening COVID-19 situation, adherence to strict lockdown measures [26,62,85], and the increasing reach of television and online news and media [62] all contributed to the heightened concerns and had a significant impact on the mental well-being of the patients. Furthermore, the unavailability of peer group support services and psychological counselling sessions [82] and the restrictions on visitors/attendants exacerbated their challenges [82].

Physical activity levels [69,87,89] were found to decrease during the pandemic due to various factors such as lack of motivation, limited time availability, and restricted access to parks, dance studios, and fitness centres. However, some individuals adapted by incorporating physical activities within their homes. In terms of dietary habits, there were observed improvements or maintenance [85,88] driven by concerns related to COVID-19. Many individuals developed a preference for home-cooked food and reduced eating out and socialising, aiming to minimise the risk of exposure to the virus.

### 3.3. Quality Appraisal of Included Studies

Among the 16 observational studies, the overall quality ranges from high to low. Specifically, 13 studies are rated as high quality, 3 studies are considered to be of moderate quality, and 1 study is assessed as low quality. Similarly, among the 21 cross-sectional surveys, the overall quality varies from high to low (Appendix **??**). Of these surveys, 11 are classified as high quality, 9 are deemed to be of moderate quality, and 2 are evaluated as low quality (Appendix **??**). In addition, there is 1 mixed-method qualitative study that incorporates both observational methodology and cross-sectional survey components. This study undergoes assessment using both tools and is determined to be of high quality.

### 4. Discussion

This review examines the impact of COVID-19 on health services and the health system's response in India during the first wave and its aftermath. It explores the initial stumbling and disruption of the system, highlighting sudden and urgent instances of imbalance, resource strain, and coordination challenges as the first objective of the study. It also sheds light on subsequent adaptations and efforts that were made with a sense of urgency to effectively address COVID-19 while at the same time striving to maintain and deliver essential health services, addressing the second objective of the study. It also reflects upon the challenges faced by healthcare providers and seekers, in relation to the third objective of the study. The review offers valuable lessons for future preparedness and identifies areas for improvement in building a resilient healthcare system capable of managing similar sudden and unprecedented crises.

This review is based on primary studies conducted in different Indian health care settings to keep the analysis based on robust research and first-hand evidence.

Highlighting the impact on the provision, utilisation, and availability of health services, consistent findings across multiple studies included in this review indicate a significant reduction in outpatient department (OPD) services [54–59,61,62,64,65,68,69,72,74,77,79,80,82,86] during the initial lockdown phase and subsequent months. Notably, among the OPD clinics at the Primary Health Care sites, noncommunicable diseases (NCDs) and immunisation clinics were particularly affected, while ANC services and general OPDs experienced relatively fewer disruptions [58]. Elective procedures across diverse healthcare areas were either completely halted or significantly scaled back, leading to delays and postponements in treatment [28,54–56,59–65,67,72–74,77]. Despite these challenges, emergency services persevered, albeit with difficulties in accessibility, transport, and resource constraints [28,54–57,59,61,62,64,67,68,72–74,77,78,80,81,84,86].

These findings align with a comprehensive survey conducted by the World Health Organisation (WHO) across 155 countries during a similar time frame, which highlighted severe interruptions in prevention and treatment services for non-communicable diseases (NCDs) as countries transitioned from sporadic cases to community transmission of the coronavirus [90].

A comprehensive systematic review conducted in multi-country settings [91] during that time further substantiates our findings, revealing a significant reduction in healthcare utilisation across various populations and countries. This reduction was particularly pronounced among individuals with less severe illnesses, in line with our observation of disrupted elective and essential services for less severe conditions, which experienced substantial declines. Conversely, emergency services for severe illnesses persisted, albeit

with necessary adaptations. Furthermore, a study [92] conducted in the United States provides additional support for these findings, demonstrating a marked decrease in medical admissions during the COVID-19 outbreak in March and April 2020 across diverse patient groups. The study also emphasises the significance of long-term research to fully understand the implications of avoiding hospitalisation during the pandemic on patients' mortality, morbidity, and quality of life, a consideration extensively discussed and observed in the included studies of this review.

In addition to evaluating the impact on health services, our study sheds light on the level of preparedness and various adaptations undertaken by healthcare facilities, addressing the second objective of the study. These adaptations encompassed changes made in precautionary and infection prevention measures [54,57,58,67,69,72,74,76,77,80], protocols and guidelines [54,59,62,64,72–74,77,81,86], staff and resource management [54,55,58,62,66–68,70,72,74,76, 77,80,82,86], PPE [27,54–59,61,62,64–66,68–70,72–74,76,78,86], preoperative/OT/post-operative measures [28,54–56,59,62,64,67,72,73,78,82,83] and infrastructure [54,58,62,66,72,77,86], demonstrating the proactive measures taken to respond to the challenges posed by the COVID-19 pandemic. These findings from our review are consistent with studies conducted in different settings and geographies during this phase of the pandemic. For instance, a study in Ireland found that healthcare infrastructures were reconfigured to facilitate the pandemic response, including the construction of structures to separate patients and the strengthening of triage systems [93]. Similarly, an analysis of datasets and literature reviews in England showed that hospitals implemented interventions to manage patient admissions and increase resource availability, including the cancellation of elective surgeries and the deployment of additional medical personnel [94]. Likewise, a National Survey conducted in the US reported comparable adaptations such as the establishment of dedicated respiratory isolation units, expanded use of inpatient telehealth, and strategies to minimise room entry [95]. These consistent findings highlight the similar efforts made to address the challenges posed by the pandemic across various healthcare settings.

Furthermore, the aforementioned survey [95], a large-scale study conducted in nursing homes across the US [96], and a prospective observational cohort study [97] conducted in both the UK and the US focusing on frontline healthcare workers, all highlighted the concerning insufficiency of personal protective equipment (PPE) and the widespread practice of reusing PPE. These findings closely align with the conclusions drawn from our review, emphasising the critical shortage of PPE and the desperate need to reuse it.

Addressing the third objective, which focuses on the impact on healthcare providers [27,54,56,58,62,65,66,68,70–72,75,76,78], a significant finding of our study is the prevalence of pandemic-related stress and burnout among healthcare providers (HCPs). HCPs, (including doctors, nurses, paramedics, and administrative staff), who cared for COVID-19 patients, feared contracting and transmitting the infection to their families. Similar findings have emerged in other studies conducted in diverse multi-country settings. For instance, an observational cohort multicentre study in acute hospital settings in the South-East of Ireland [98] highlighted the scarcity of research focusing on mental health issues during the pandemic and found a notable prevalence of psychological distress among HCPs in those settings. On similar lines, a systematic review during the time in selected Asian countries [99] examined viral epidemic outbreak studies and concluded that the prevalence of anxiety, depression, acute and PTSD, and burnout was high during and after outbreaks; with problems having a long-lasting effect on the mental health of HCPs.

Furthermore, our study observed instances of stigmatisation against HCPs, which is consistent with the statement released by the International Committee of the Red Cross (ICRC) [100], reporting over 600 incidents of violence, harassment, or stigmatisation targeting healthcare providers, patients, and medical infrastructure in relation to the COVID-19 pandemic.

While the review highlights the research and evidence-based picture of the challenges posed by the pandemic and the response to it, it is crucial to acknowledge the inherent limitations within the primary studies included. These studies were conducted during the early stages of the pandemic, necessitating careful consideration of the spatial and

temporal aspects of the findings. Extrapolating the review's findings to a different time period or phase of the pandemic may contradict them and limit their generalisability. Furthermore, the contextual nature of the findings within Indian settings prevents their direct applicability to other countries.

The majority of the studies aimed to assess changes in the status of health services across different health conditions and levels, considering the immense and sudden pressures faced by the health system. Due to the rapidly evolving situation, limited attention was given to meticulous sample selection methods and research study design. Consequently, all the studies relied on cross-sectional surveys with purposefully selected samples. While this approach may have compromised the robustness of the evidence collected, it effectively captured the sudden impact of the pandemic on various health services, especially when timely evidence was crucial for interventions. Additionally, the study faced other limitations, including the inability to conduct a meta-analysis due to significant heterogeneity in the included studies, the potential exclusion of important data sources such as studies with smaller sample sizes, reports, and subjectivity in our outcome assessments.

## 5. Conclusions

In conclusion, this study sheds light on the profound and sudden impact of COVID-19 on health services during the initial wave, resulting in disruptive effects on OPD and elective services and raising concerns about suboptimal emergency care. The healthcare system promptly responded by implementing quick and possible adaptations in staff management, resource allocation, and infection prevention. These valuable findings and insights provide essential knowledge to enhance our understanding of the necessary measures, approaches, and level of preparedness required to build resilient health systems. Incorporating these insights into future strategies will ensure that the health system is well-prepared and resilient in effectively addressing similar crises proactively.

Furthermore, the study underscores the significant toll on the mental health and well-being of healthcare providers, who endured unparalleled challenges and resource constraints. It also illuminates the widespread concerns and fears amongst healthcare seekers, particularly regarding the worsening of their underlying diseases or conditions due to significant delays, formidable barriers, and suboptimal services for diagnosis, treatment, and follow-up.

To mount an effective response to future similar emergencies, it is imperative to prioritise the readiness and preparedness of the health system. This involves adopting agile and efficient strategies to strengthen healthcare infrastructure, optimise resource allocation, and implement comprehensive protocols. Addressing the concerns of healthcare seekers during emergencies and implementing measures to support healthcare providers are vital. By placing emphasis on these aspects, we can proactively enhance the resilience of the health system, ensuring improved outcomes in times of crisis.

**Supplementary Materials:** The protocol registered on PROSPERO can be downloaded at: https://www.crd.york.ac.uk/prospero/display_record.php?RecordID=227327 (accessed on 24 June 2023).

**Author Contributions:** Conceptualisation, A.S.C., K.S., R.B. and S.K.; methodology, A.S.C., K.S., R.B. and S.K. software, A.S.C. and A.N.; validation, K.S., R.B. and S.K.; formal analysis, A.S.C. and A.N.; investigation, A.S.C. and A.N.; resources, K.S., R.B. and S.K.; data curation, A.S.C. and A.N.; writing—original draft preparation, A.S.C.; writing—review and editing, A.S.C., K.S., R.B. and S.K.; visualisation, A.S.C. and A.N.; supervision, A.S.C., K.S., R.B. and S.K.; project administration, A.S.C., K.S., R.B. and S.K.; funding acquisition, A.S.C. and K.S. All authors have read and agreed to the published version of the manuscript.

**Funding:** This research was funded by Asian Development Bank (47354-003).

**Institutional Review Board Statement:** This review is a part of research approved by Human Research Ethics Committee-Ethics committee of Delhi—Sigma-IRB (Institutional Review Board)-IRB Approved date/number—14.12.2020/10044/IRB/D/20-21.

**Informed Consent Statement:** Not applicable.

**Data Availability Statement:** The data used in this review are derived from published literature and publicly available sources. No additional data sets were generated or analysed specifically for this review. All references and sources cited in this review are provided for transparency and accessibility.

**Conflicts of Interest:** The authors declare no conflict of interest.

# Appendix A

*Appendix A.1. Quality Appraisal of Observational Studies Using Adapted CASP Checklist*

| S.No. | Studies | Was There a Clear Statement of the Aims? | Is a Qualitative Methodology Appropriate? | Was the Research Design Appropriate to Address the Research Aims? | Was the Recruitment Strategy/Sampling Appropriate to the Aims of the Research? | Was the Data Collected in a Way That Addressed the Research Issue? | Has the Relationship between Researcher and Participant been Adequately Considered? Research issue? | Have Ethical Issues been Taken into Consideration? | Was the Data Analysis Sufficiently Rigorous? | Is There a Clear Statement of Findings? | Do the Conclusions Drawn in the Research Report Flow form the Analysis or Interpretation of the Data? | Quality Assessment |
|---|---|---|---|---|---|---|---|---|---|---|---|---|
| 1 | Goyal N et al. [55] | x | x | x | x | x | x | x | x | x | x | High |
| 2 | Singh M et al. [57] | x | x | x | x | x | | | x | x | x | Medium |
| 3 | Choudhary R et al. [60] | x | x | x | x | x | | | x | x | x | High |
| 4 | Goyal M et al. [61] | x | x | x | x | x | x | x | x | x | x | High |
| 5 | Deshmukh, S [62] | x | x | x | x | x | | | x | x | x | Medium |
| 6 | Gupta A et al. [64] | x | x | x | x | x | | x | x | x | x | High |
| 7 | George CE et al. [66] | x | x | x | x | x | x | x | x | x | x | High |
| 8 | Agarwal D et al. [67] | x | x | x | x | x | | x | x | x | x | High |
| 9 | Khurana DK et al. [69] | x | x | x | x | x | | x | | x | x | Low |
| 10 | Babu N et al. [74] | x | x | x | x | x | | x | x | x | x | High |
| 11 | Mahajan NN et al. [77] | x | x | x | x | x | x | x | x | x | x | High |
| 12 | Verma A et al. [79] | x | x | x | x | x | x | x | x | x | x | High |

| S.No. | Studies | Was There a Clear Statement of the Aims? | Is a Qualitative Methodology Appropriate? | Was the Research Design Appropriate to Address the Research Aims? | Was the Recruitment Strategy/Sampling Appropriate to the Aims of the Research? | Was the Data Collected in a Way That Addressed the Research Issue? | Has the Relationship between Researcher and Participant been Adequately Considered? Research issue? | Have Ethical Issues been Taken into Consideration? | Was the Data Analysis Sufficiently Rigorous? | Is There a Clear Statement of Findings? | Do the Conclusions Drawn in the Research Report Flow form the Analysis or Interpretation of the Data? | Quality Assessment |
|---|---|---|---|---|---|---|---|---|---|---|---|---|
| 13 | Jain A et al. [81] | x | x | x | x | x | x | x | x | x | x | High |
| 14 | Ghosh J et al. [83] | x | x | x | x | x | x | x | x | x | x | High |
| 15 | Panda PK et al. [84] | x | x | x | x | x | x | x | x | x | x | High |
| 16 | Ghosal S et al. [87] | x | x | x | x | x | x | x | x | x | x | High |
| 17 | Gautam P et al. [54] | x | | | | | | | x | x | x | Medium |

*Appendix A.2. Quality Appraisal of Cross-Sectional Surveys Using Adapted JBI Tool*

| S.No. | | Were the Aims/Objectives of the Study Clear? | Was the Study Design Appropriate for the Stated Aim(s)? | Was the Sample Size Justified? | Was the Target/Reference Population Clearly Defined? (Is It Clear Who the Research Was About?) | Was the Sample Frame Taken from an Appropriate Population Base So That It Closely Represented the Target/Reference Population under Investigation? | Was the Selection Process Likely to Select Subjects/Participants That Were Representative of the Target/Reference Population under Investigation? | Were Measures Undertaken to Address and Categorise Non-Responders? | Were the Risk Factor and Outcome Variables Measured Appropriate to the Aims of the Study? (e.g. Could You See the Questionnaire?) | Were the Risk Factor and Outcome Variables Measured Correctly Using Instruments/Measurements That Had been Trialled, Piloted or Published previously? | Is It Clear What Was Used to Determined Statistical Significance and/or Precision Estimates? (e.g., p Values, CIs) | Were the Methods (Including Statistical Methods) Sufficiently Described to Enable Them to be Repeated? | Were the Basic Data Adequately Described? | Does the Response Rate Raise Concerns about Non-Response bias? | If Appropriate, Was Information about Non-Responders Described? | Were the Results Internally Consistent? | Were the Results for the Analyses Described in the Methods, Presented? | Were the Authors' Discussions and Conclusions Justified by the Results? | Were the Limitations of the Study Discussed? | Were There Any Funding Sources or Conflicts of Interest That May Affect the Authors' Interpretation of the Results? | Was Ethical Approval or Consent of Participants Attained? | Overall Assessment |
|---|---|---|---|---|---|---|---|---|---|---|---|---|---|---|---|---|---|---|---|---|---|---|
| 1 | Nair A et al. [56] | x | x | x | x | x | x | | x | x | x | x | x | | | x | x | x | x | x | | Medium |
| 2 | Garg S et al. [58] | x | x | x | | | x | | x | x | x | x | x | | | x | x | x | x | x | x | Medium |
| 3 | Nasta AM et al. [59] | x | | x | x | x | | | x | x | x | x | x | | | x | x | x | x | x | | Medium |
| 4 | Keshav K et al. [63] | x | x | x | x | x | x | | x | x | x | x | x | | | x | x | x | x | x | x | High |
| 5 | Sahu D et al. [65] | x | x | x | x | x | x | x | x | x | x | x | x | | | x | x | x | x | x | x | High |
| 6 | George CE et al. [66] | x | x | x | x | x | x | x | x | x | x | x | x | | | x | x | x | x | x | x | High |
| 7 | Barik S et al. [68] | x | x | x | x | x | x | x | x | x | x | x | x | | | x | x | x | x | x | x | High |
| 8 | Wilson W et al. [27] | x | x | x | x | x | x | x | x | x | x | x | x | | | x | x | x | x | x | x | High |
| 9 | Khasne RW et al. [70] | x | x | x | x | x | x | x | x | x | x | x | | | | x | x | x | x | x | x | High |
| 10 | Das A et al. [71] | x | x | x | x | x | x | x | x | x | x | x | | | | x | x | x | x | x | x | Medium |
| 11 | Venkataram T et al. [72] | x | x | x | x | x | x | x | x | x | x | x | | | | x | x | x | x | x | x | High |
| 12 | Bhandoria G et al. [73] | x | | x | x | x | x | | x | x | | x | x | | | x | x | x | x | x | | Low |
| 13 | Khanna RC et al. [75] | x | x | x | x | x | | | x | x | x | x | x | | | x | x | x | x | x | x | High |
| 14 | Chatterjee SS et al. [76] | x | x | x | x | x | x | x | x | x | x | x | x | | | x | x | x | x | x | x | Medium |

| S.No. | Study | Were the Aims/Objectives of the Study Clear? | Was the Study Design Appropriate for the Stated Aim(s)? | Was the Sample Size Justified? | Was the Target/Reference Population Clearly Defined? (Is It Clear Who the Research Was About?) | Was the Sample Frame Taken from an Appropriate Population Base So That It Closely Represented the Target/Reference Population under Investigation? | Was the Selection Process Likely to Select Subjects/Participants That Were Representative of the Target/Reference Population under Investigation? | Were Measures Undertaken to Address and Categorise Non-Responders? | Were the Risk Factor and Outcome Variables Measured Appropriate to the Aims of the Study? (e.g. Could You See the Questionnaire?) | Were the Risk Factor and Outcome Variables Measured Correctly Using Instruments/Measurements That Had been Trialled, Piloted or Published previously? | Is It Clear What Was Used to Determined Statistical Significance and/or Precision Estimates? (e.g., p Values, CIs) | Were the Methods (Including Statistical Methods) Sufficiently Described to Enable Them to be Repeated? | Were the Basic Data Adequately Described? | Does the Response Rate Raise Concerns about Non-Response bias? | If Appropriate, Was Information about Non-Responders Described? | Were the Results Internally Consistent? | Were the Results for the Analyses Described in the Methods, Presented? | Were the Authors' Discussions and Conclusions Justified by the Results? | Were the Limitations of the Study Discussed? | Were There Any Funding Sources or Conflicts of Interest That May Affect the Authors' Interpretation of the Results? | Was Ethical Approval or Consent of Participants Attained? | Overall Assessment |
|---|---|---|---|---|---|---|---|---|---|---|---|---|---|---|---|---|---|---|---|---|---|---|
| 15 | Subbian A et al. [28] | x | x | x | x | x | x | | x | x | x | x | x | x | | x | x | x | x | x | x | High |
| 16 | Moorthy RK et al. [78] | x | x | x | x | x | x | | x | x | x | x | x | x | | x | x | x | x | x | | Medium |
| 17 | Joshi R et al. [80] | x | x | x | x | x | x | | x | x | x | x | x | x | | x | x | x | x | x | | Low |
| 18 | Mitra M et al. [82] | x | x | x | x | x | x | | x | x | x | x | x | x | | x | x | x | x | x | x | Medium |
| 19 | Nachimuthu S et al. [85] | x | x | x | x | x | x | x | x | x | x | x | x | x | | x | x | x | x | x | x | High |
| 20 | Prasad N et al. [86] | x | x | x | x | x | x | | x | x | x | x | x | x | | x | x | x | x | x | x | Medium |
| 21 | Chopra S et al. [88] | x | x | x | x | x | x | | x | x | x | x | x | x | | x | x | x | x | x | x | High |
| 22 | Nilima N et al. [26] | x | x | x | x | x | x | | x | x | x | x | x | x | | x | x | x | x | x | x | Medium |

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
