# Peer review of "The Health System’s Response to and the Impact of COVID-19 on Health Services, Providers, and Seekers: A Rapid Review in the Wake of the Pandemic"

_covid, doi:10.3390/covid3080081_

Round 1

Reviewer 1 Report

The authors reviewed the impact of COVID-19 on health services, providers, and seekers. Overall, the study is meaningful and of good merits, and may enhance the resilience and preparedness of healthcare systems for future challenges. Some concerns as follows:

1. As the author mentioned several primary studies have examined the impact of COVID-19 on the health system and its stakeholders (lines 48-49), I suspect related references should be cited. In the meantime, the authors should imply what they have already mentioned, and compare your review with previous findings.

2. The authors preferentially chose studies conducted exclusively within the Indian context. Why?

Maybe the COVID-19 situation as well as health care settings in India should be clearly mentioned in the Background. Also, whether the background in India different from other countries?

Reviewer 2 Report

This is a well-thought out and well-executed review of the literature regarding three objectives with respect to India:

a. Assess the impact of COVID-19 on the provision, utilisation, and availability of health services

b. Understand the health system’s response – adaptations, interventions and efforts for continuity and resumption of services 

c. Evaluate the implications of COVID-19 and its response on individuals – Health care providers and health care seekers.

The strengths of this study are the extent of the review conducted and the detailed analysis of the returns from a number of important perspectives. Furthermore, the review is very clear and well written. The weaknesses are that insufficient details are provided regarding the methods undertaken by the authors. Although the methods are each mentioned, some are not described in sufficient detail nor referenced for the review to be replicated. Another limitation is the PRISMA diagram that is not as clear in its representation of the work undertaken as the written account. Still each of these concerns is likely easily corrected by these authors as this is a commendable piece of research.

Line by line suggested edits.

12-37 The authors should note that their abstract is 325 words. The word limit for abstracts for COVID is 200. Please adjust the length of the Abstract accordingly.

38 Please add “COVID-19” as the first of the keywords.

42-47 There are three sentences in this paragraph. Each makes a claim that must be substantiated with references to peer reviewed research.

48-50 The authors mention “several primary studies” yet have failed to cite these studies. They must be cited and included in the references.

54-60 The authors are asked to please make it clear that these objectives relate to India.

64 Where is the University of York located? Please add this information.

67-98 In relation to the inclusion and exclusion criteria, the authors are asked to specify the exact keywords they used to do their searches. These can either be noted in section 2.3. or following each of the inclusion and exclusion criteria in section 2.1. and 2.2.

100 Please indicate why the Mendeley reference management software was selected and provide a current reference to other research using this reference manager in this way.

102 Please specific why the authors selected Covidence as the review software and provide a reference to demonstrate its similar, recent use by other researchers.

107-108 Were there disagreements? If so, please indicate the type of disagreements and how the disagreements were resolved through discussion and when discussion wasn’t enough and an independent reviewer was needed.

110-111 How did the screened articles undergo a further relevance check? Please explain.

113 Why was the particular pilot-tested data collection form of Excel selected? Please explain and provide a reference for similar, recent use of this form by other researchers.

115-120 In mentioning these data items, please refer back to the author’s stated objectives and indicate how the organization of the data relates to these objectives.

122 Please indicate why they authors anticipated heterogeneity in this regard.

124 In providing this outline of primary and secondary outcomes to follow, please relate how these were related to the authors’ objectives provided in lines 55-60.

151-152 Why were the CASP and AXIS selected for use? Please explain and include references to recent peer reviewed research demonstrating the use of these tools by other researchers.

156 Were there any discrepancies? If so, of what type were they and how was consensus reached?

161 Please provide the authors’ definition for “narrative synthesis” and cite a recent peer reviewed reference that defines the term in the same way.

165 Please provide details on “this method”. Without any details, this part of the study can’t be reproduced.

173 If the authors have retained the information, it would be helpful to know whether the duplicates were purely in relation to using five databases or if any individual data base contained duplicates.

178-202 The authors are asked to be consistent with the use of “n”. In “Identification”, on the left side, “n” is used only in relation to “n = 1477” but not for the individual data bases. As well, “n” is not used on the right side of “Included”. Please use “n” in relation to all.

178-182 Please reorganize the information in the left box so that the returns from each of the data bases is on its own line. It is too difficult to read as it is displayed now.

181-202 The authors have used blue to number both those records included (except for in the top left box)and those excluded. Please differentiate the included from the excluded records by using blue only for the included or the excluded, not both.

191-202 The two boxes on the right should be combined into one box as they represent the same operation. In doing so, change “excluded, with reasons” to “excluded.”

205-212 Please mention if either the setting or type of participant affected the choice of study conducted.

212 Please move Table 1 to after this paragraph, it is too far away from the first mention of Table 1 in 204. As well, provide information regarding the various sub-subsections to follow and their relationship to the three objectives of the study provided by the authors in lines 55-60.

216-229 Please change the display of these lines from left justification to full justification.

251 Please spell out STEMI in full before using the acronym.

307 Delete “general”.

409-428 The paragraph from 419-428 is a repeat of the paragraph from 409-418. Please delete the second paragraph of the two.

480 Given that all of the information reported in these section regards Table 1, there is no need to mention Table 1 here. Delete “(Table 1)”.

520-521 In the heading of Table 1, please indicate the categorization of each of the columns and the number of studies reported. For the first study, please line up “1.” with the rest of the column.

522-525 This information has been provided already in lines 150-152 and need not be repeated here.

529-531 To correspond with the title of the tables provided in lines 676-679 to follow, Change “Appendix 1a” to Appendix 1A” and change “Appendix 1b” to Appendix 1B”.

536-630 During this Discussion, please return to the three objectives outlined in lines 55-60 and discuss the research in relation to these three objectives.

547 Please cite the multiple studies.

549-551 Please cite the appropriate studies in relation to this claim.

551-552 Please cite the appropriate studies in relation to this claim.

650-652 Please delete this statement regarding public health campaigns as nowhere in this review have the authors investigated public health campaigns.

676-680 Information on tables is expected to be centred, not left justified. Please correct.
